# The incidence of candidate binding sites for β-arrestin in *Drosophila* neuropeptide GPCRs

**Paul H. Taghert** *

Department of Neuroscience, Washington University School of Medicine, St. Louis, MO, United States of America

* taghertp@wustl.edu

## Abstract

To support studies of neuropeptide neuromodulation, I have studied beta-arrestin binding sites (BBS's) by evaluating the incidence of BBS sequences among the C terminal tails (CTs) of each of the 49 *Drosophila melanogaster* neuropeptide GPCRs. BBS were identified by matches with a prediction derived from structural analysis of rhodopsin:arrestin and vasopressin receptor: arrestin complexes [1]. To increase the rigor of the identification, I determined the conservation of BBS sequences between two long-diverged species *D. melanogaster* and *D. virilis*. There is great diversity in the profile of BBS's in this group of GPCRs. I present evidence for conserved BBS's in a majority of the *Drosophila* neuropeptide GPCRs; notably some have no conserved BBS sequences. In addition, certain GPCRs display numerous conserved compound BBS's, and many GPCRs display BBS-like sequences in their intracellular loop (ICL) domains as well. Finally, 20 of the neuropeptide GPCRs are expressed as protein isoforms that vary in their CT domains. BBS profiles are typically different across related isoforms suggesting a need to diversify and regulate the extent and nature of GPCR:arrestin interactions. This work provides the initial basis to initiate future *in vivo*, genetic analyses in *Drosophila* to evaluate the roles of arrestins in neuropeptide GPCR desensitization, trafficking and signaling.

## Introduction

Neuropeptides and neurohormones signal to target cells via specific interactions with membrane proteins. The large majority of receptors for neuropeptides and peptide hormones are G protein-coupled receptors (GPCRs) that transmit the modulatory signals via second messenger pathways [2–5]. GPCR signaling must also be down-regulated in order to provide opportunity for the next round of signals, whether that be minutes, hours or days away. There are two large categories of receptor de-sensitization, and both feature GPCR phosphorylation. Homologous de-sensitization reflects phosphorylation of agonist-bound GPCR: such activated GPCRs present conformations that promote phosphorylation by one or more G protein-coupled receptor kinases (GRKs) and that in turn recruit association with arrestin proteins [6, 7]. The arrestins uncouple GPCRs from heterotrimeric G proteins and cluster the phosphorylated GPCRs into clathrin-coated pits [8, 9], leading either to clearance from the membrane via endocytosis, or eventual GPCR re-cycling for further signaling episodes. The second category of GPCR de-

**Data Availability Statement:** All relevant data are within the paper and its Supporting Information files.

**Funding:** The work was made possible by grant support to PHT from the NIH - NINDS R01

NS108393-20 (https://www.ninds.nih.gov/). The funders played no role in the study design, data collection and analysis, decision to publish or preparation of the manuscript.

**Competing interests:** The author has declared that no competing interests exist.

sensitization is termed Heterologous and it describes GPCR phosphorylation via protein kinase A (PKA) or protein kinase C (PKC) in response to activation of a heterologous receptor [10].

Historically, arrestin biology traces its roots to pioneering work on the desensitization of phosphorylated rhodopsin: this established the key role of the S48 protein in promoting the cessation of rhodopsin GPCR signaling and led to naming the S48 protein "arrestin" [11, 12]. The name was later refined to visual arrestin, because two related proteins were found to promote the desensitization of many neurotransmitter and neuropeptide GPCRs; the latter group was termed the non-visual-arrestins, or β-arrestin-1 and -2 [13–16]. The fate of the GPCR: β-arrestin complex can vary according to receptor [17] and the affinity of β-arrestins for phosphorylated GPCRs varies, with evidence for at least two categories [18, 19]. Strong affinity is correlated with sustained interactions and endocytosis of the GPCR; weaker affinity is thought to reflect early dissociation with GPCR, permitting prompt recycling to the plasma membrane.

In fact, the roles of arrestins in regulating GPCR signaling are not limited to signal termination, and remain at least in part enigmatic. There is strong consensus that β-arrestins serve critical scaffold functions to recruit additional GPCR interactors and additional, distinct rounds of GPCR signaling within endosomes that are G protein-independent [20, 21]. These and other findings have generated a theory of 'biased agonism', such that selective ligands will predispose a single GPCR towards either a G protein-dependent signaling mode, or instead a β-arrestin-dependent one [22, 23]. *Drosophila* express a single β-arrestin orthologue named *kurtz* [24]. With functional expression *in vitro*, mammalian β-arrestins:GFP is normally resident in the cytoplasm, but is recruited to the membrane when a co-expressed neuropeptide GPCR is activated by its cognate ligand [25]; *Kurtz*:GFP behaves similarly [26]. Several different efforts have shown the significance of Kurtz *in vivo* in regulating a host of receptors that underlie critical developmental signaling. These include the Toll [27, 28], the hedgehog [29] and the Notch [30] signaling pathways. However, at present the relationship between Kurtz and neuropeptide GPCR signaling in *Drosophila in vivo* remains poorly described and understudied.

To promote renewed consideration of the roles of β-arrestins in neuropeptide GPCR physiology in *Drosophila*, here I present a systematic estimation of the incidence and details of potential β-arrestin binding sites on all 49 annotated *Drosophila* neuropeptide GPCRs. The predicate for this effort is a recent proposal and evaluation of a candidate β-arrestin binding sequence (BBS) on the C termini of GPCRs [1]. That study was based on parallel, structural analyses of a rhodopsin::visual-arrestin complex and of a vasopressin receptor::β-arrestin complex.

## Materials and methods

### Gene selection

I used a compilation of 49 *Drosophila melanogaster* neuropeptide and protein hormone GPCRs provided by Flybase (http://flybase.org/). These were divided into two lists: (i) CLASS A GPCRs (http://flybase.org/reports/FBgg0000041), termed, "CLASS A GPCR NEUROPEPTIDE AND PROTEIN HORMONE RECEPTORS" and (ii) CLASS B GPCRs (http://flybase.org/reports/FBgg0000099), termed "CLASS B GPCRs, SUBFAMILY B1".

### GPCR structure

I assigned TM domains based on predictions obtained in the GenPep reports for individual GPCRs (S1 Text "Collated and annotated *D. melanogaster* and *D. virilis* GPCR sequences"). When such predictions were lacking or problematic, I consulted on-line prediction services:

*services.healthtech.dtu.dk/service.php?TMHMM-2.0*. Genbank Reference numbers for individual GPCRs in *D. melanogaster* are listed in S1 Table, and for both *D. melanogaster* and *D. virilis* in S1 Text.

### Blast searches

I performed Blastp searches (blast.ncbi.nlm.nih.gov/Blast.cgi?PAGE=Proteins) using each of the 49 *D. melanogaster* GPCRs as queries with the blastp algorithm. I retrieved *D. virilis* (taxid 7244) proteins that carried E values < -100. I judged orthologous BBS sites by proximity (< 20 amino acid (AA) distance, as bounded by the end of the 7th TM domain and the end of the CT), and by the sequence of the BBS; I sought confirmation by inspection of sequences proximal and distal to candidate BBS sites.

### Gene annotations

Several *D. melanogaster* GPCRs exhibit multiple protein isoforms (S1 Table). If these revealed a difference in BBS profile for a single GPCR, I searched for pertinent orthologous isoforms in other *Drosophila* species by blastp searches. If they were not retrieved by blastp, I inspected genomic sequences directly using NCBI Reference Sequence models (e.g., https://www.ncbi.nlm.nih.gov/nuccore/1829008042?report=graph&v=9255109:9263757). Using the predictions of the *D. melanogaster* gene structures, I produced conceptual translations of candidate genomic sequences, and searched for matches to isoforms of interest. I inspected and produced *de novo* annotations for the following eight GPCR isoforms in one or more non-*melanogaster* species: AstA-R2-PB, CAPA-R, CCAP-R, CCHa-R2 PA, Moody PC, RYa-R PB, Tk-R 86C, and Trissin-R. The data and analyses documenting these annotations are presented in S2–S8 Texts documents entitled: de novo [GPCR] isoform annotations". To evaluate the validity of a comparison between just *D. melanogaster* and *D. virilis*, I extended the number of species for comparison to 19 for certain GPCRs. The data and analyses documenting these annotations and subsequent alignments are presented in S9–S20 Texts documents entitled: or "Multi-species analysis of [GPCR]".

## Results

### Overview

I collected predicted translations for each of the 49 *Drosophila* neuropeptide GPCRs, as categorized by flybase ((http://flybase.org/). 44 are classified as Family A (rhodopsin-like) receptors and five as Family B (secretin receptor-like) receptors. I used the Flybase annotation system to include different predicted isoforms (e.g., PA, PB, etc), and focused on sequence features in the GPCR intracellular domains: the three Intracellular loops (ICL1-3) and the C-terminal (CT) domains. S1 Table lists the 49 genes included in this study, including 21 for which Flybase annotations indicate the occurrence of multiple C termini. From inspection, 14 genes generate multiple protein isoforms by alternative splicing. In addition, eight do so by a mechanism of stop suppression, according to the postulate of Jungreis *et al.* [31]: this is a feature of numerous, diverse genes, including many in *Drosophila*. One *Drosophila* neuropeptide GPCR gene (*CCHa-R2*) appears to use both mechanisms. One (*CAPA-R*) produces an isoform by alternative splicing that is truncated within TM6. I manually scanned the intracellular domains of the unique isoforms of all predicted neuropeptide GPCR sequences for precise matches to the predicted β-arrestin2 binding site (BBS), as defined by Zhou *et al.* [1]. The BBS may appear in a "short" form, [S/T-(X1)-S/T-(X2)-(X3)-(S/T/E/D)], where X1 = any AA and X2 and X3 = any AA, except proline. A "long" BBS form was also proposed, by which the second and

third S/T residues are separated by a second X, with no restricted AAs. Precise matches to either the long or short form of a BBS correspond to a "complete" site as defined by Zhou *et al.* [1]. That is in contrast to what they term a "partial" site, one that contains some but not all elements of the predicted BBS code sequence. Here I have focused exclusively on precise (i.e., complete) BBS matches. Zhou *et al.* [1] concentrated on BBS's located within the CT domains, based on their structural analysis of GPCR::arrestin interactions. By these criteria, and as described below in greater detail, 41 of 49 *Drosophila melanogaster* GPCRs contain at least one isoform with at least one perfect candidate BBS in a CT domain. Of the eight receptors that contained no precise matches in the CT domains, two contain precise matches in their ICLs.

To increase the rigor by which candidate BBS's are categorized for this GPCR group, I extended the analysis to a second *Drosophila* species–*D. virilis*. Whereas *D. melanogaster* is a member of the *Sophophora* sub-genus, *D. virilis* is a member of the *Drosophila* one. These two sub-genera are estimated to have diverged at least 60 Myr ago [32]. The demonstration that *D. melanogaster* and *D. virilis* display sequence conservation within the coding regions of a particular gene, or within the putative regulatory regions, is considered a useful indicator of functional conservation of protein or *cis*-regulatory sequences [e.g., 33–35]. As applied, this comparison reduced the occurrence of 'putatively-functional' arrestin binding sites such that only 36 of the 49 GPCR CTs contain precise BBS candidates that could be matched across the two species by position and/ or sequence.

To gain a general perspective, I measured three parameters concerning the sequence properties and distributions of the candidate arrestin binding sites along the CT lengths of the 49 neuropeptide GPCRs (Fig 1). Firstly, neuropeptide GPCRs in both species typically had 1 to 2 conserved binding sites, with a range extending from 0 to as many as 7 (Fig 1A). Secondly, the majority of BBS's were either 6 or 7 AAs in length (Fig 1B), matching the model predicted by Zhou *et al.* [1]. About 1/3 of the BBS's were longer (>10 AA) and contained multiple BBS matches (these represent compound sites which, as noted by Zhou *et al.* [1] occurs in the rhodopsin CT, and these increased the average number of AAs per BBS to 9.0 +/- 0.5 (n = 58) for *melanogaster* neuropeptide GPCRs and 8.6 +/- 0.4 (n = 59) for the *virilis* ones. The longest conserved BBS found among these GPCRs spanned 20 AA (Fig 1B). Finally, I asked whether there might be a positional bias for the incidence of candidate BBS's along the length of a neuropeptide GCPR CT. Candidate BBS's appeared equally likely to occur at all positions along the C termini of GPCRs, with the exception of the first 10% of the length immediately following the 7th TM (Fig 1C).

Direct cryo-EM observations reveal details of β-arrestin interactions with the receptor core [36]. There is strong evidence that β-arrestin interacts with GPCRs based on conformational features presented several distinct regions, including intracellular domains that are independent of the CT region [37]. Although Zhou *et al.* [1] focused their attention on BBS's only in CT domains, there are many such sites, conserved between the two species, present in the ICLs of these 49 neuropeptide GPCRs. I call these "BBS-like" sites because they match the sequence prediction for BBS's defined by Zhou *et al.* [1], but are found in intracellular domains distinct from the CT. Fig 2 presents an overview of these sequences. 19 of 49 *Drosophila* neuropeptide GPCRs contain one or more conserved BBS-like sequences within ICL domains, hence the large majority of this receptor set do not contain any (Fig 2A). The length of ICL BBS-like sequences ranged from six to 13 for *D. melanogaster* and six to 32 for *D. virilis* (Fig 2B). The reason for the discrepancy was a single example (CCKL-R 17D3) in which a single long BBS-like in *D. virilis* is divided by two non-complying AAs in *D. melanogaster*. Based on their proximity and over-all sequence similarities, I rated these two nearly-contiguous BBS sites in *D. melanogaster* as correspondent to the single one in *D. virilism*: these are detailed in later Figures. The average BBS-like length was close to the canonical BBS prediction of 6 to 7 AA's [1]:

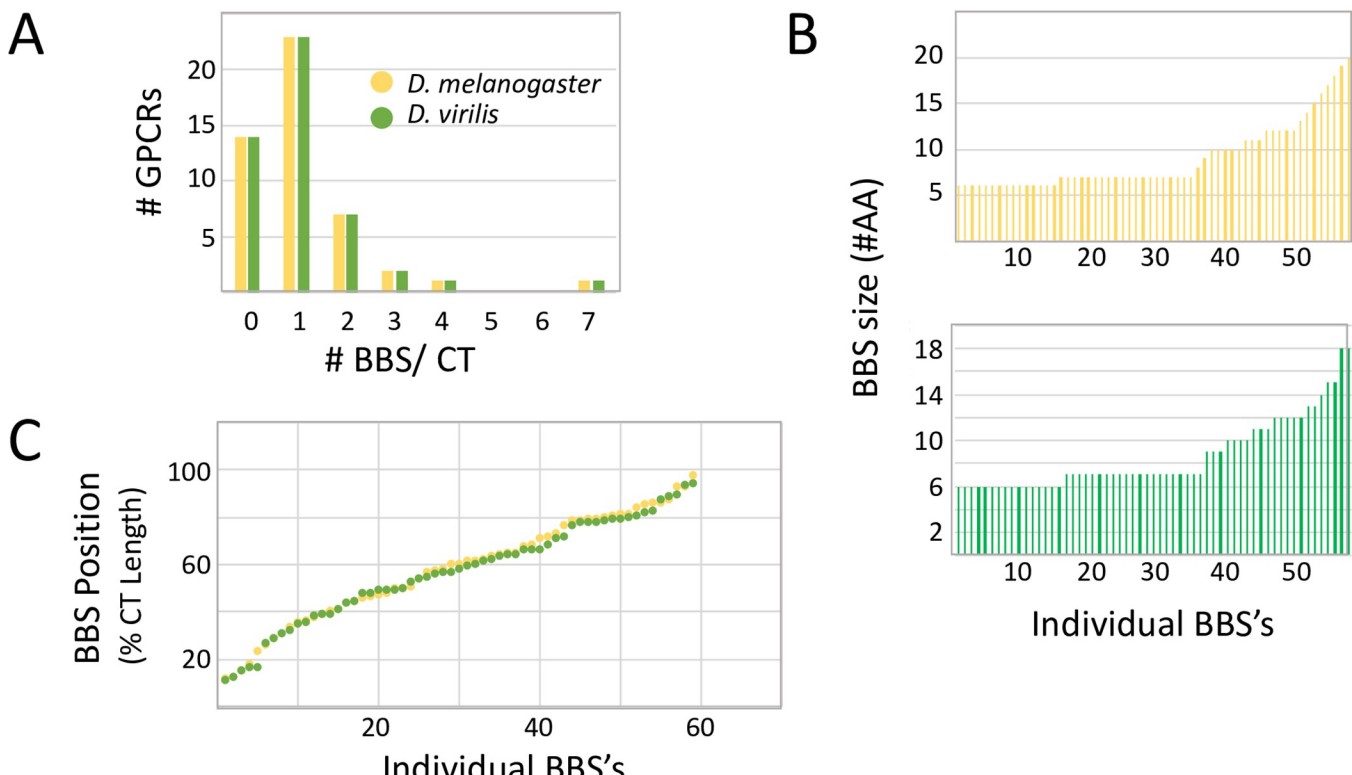

**Fig 1. A numerical overview of the incidence of BBS sequences in the CT domains of 49 *Drosophila* neuropeptide GPCRs.** (A). A plot of the number of BBS per GPCR. The equal numbers across the range between *D. melanogaster* and *D. virilis* is explained by the applied definition of a BBS, based on conservation between these two species. (B). The distribution of BBS size across all occurrences for *D. melanogaster* (yellow) and *D. virilis* (green) genes. The majority correspond to the canonical 6–7 AA size, although a sizable number exhibit greater length. (C). The distribution of BBS position across the length of the CT domain: the plot reveals comparable representation at all positions, with the exception of the very first 10% immediately following the 7th TM.

7.6 +/- 0.6 (SEM) AAs in *D. melanogaster* (n = 25) and 8.0 +/- 1.1 (SEM) AAs in *D. virilis* (n = 25). Lastly, among ICL domains, BBS-like sequences were most strongly represented in ICL3 (Fig 2C). Of the 19 GPCRs containing BBS-like sequences, 17 contained one or more in ICL3: three of these 17 receptors also contained them in ICL1, and two also contained them in ICL2. Three other GPCRs contained conserved BBS-like sequences in ICL2 only.

Conserved BBS-like sequences in ICL2 are potentially notable based on studies demonstrating ICL2 stabilization of β-arrestin2: receptor core interactions, by a mechanism that is independent of arrestin binding to the CT [38]. I found five neuropeptide GPCRs with conserved BBS-like sequences in the ICL2 (S1 Fig), just downstream (2–5 residues) of a critical Proline that is situated at the (DRY)+6 position. These included the AstA-R1, the paralogous receptors CCKL-R 17D1 and -R 17D3 and the paralogous receptors PK2-R1 and -R2. AstA-R1 also has paralogue receptor, AstA-R2, but the latter does not contain an ICL2 BBS-like sequence. As a side note, I found evidence for four rhodopsin-like *Drosophila* neuropeptide GPCRs that lack the conserved Proline at (DRY)+6 (S2 Fig). In this regard this neuropeptide GPCR set is reminiscent of the chemokine receptor family which largely express Alanine at the (DRY)+6 position [38].

## Detailing the evolutionary comparisons for the 49 *Drosophila* neuropeptide GPCRs

The occurrence and sequences of conserved BBS's found among the 49 *Drosophila* neuropeptide GPCRs are illustrated in schematic fashion in Figs 3–11. These representations are based

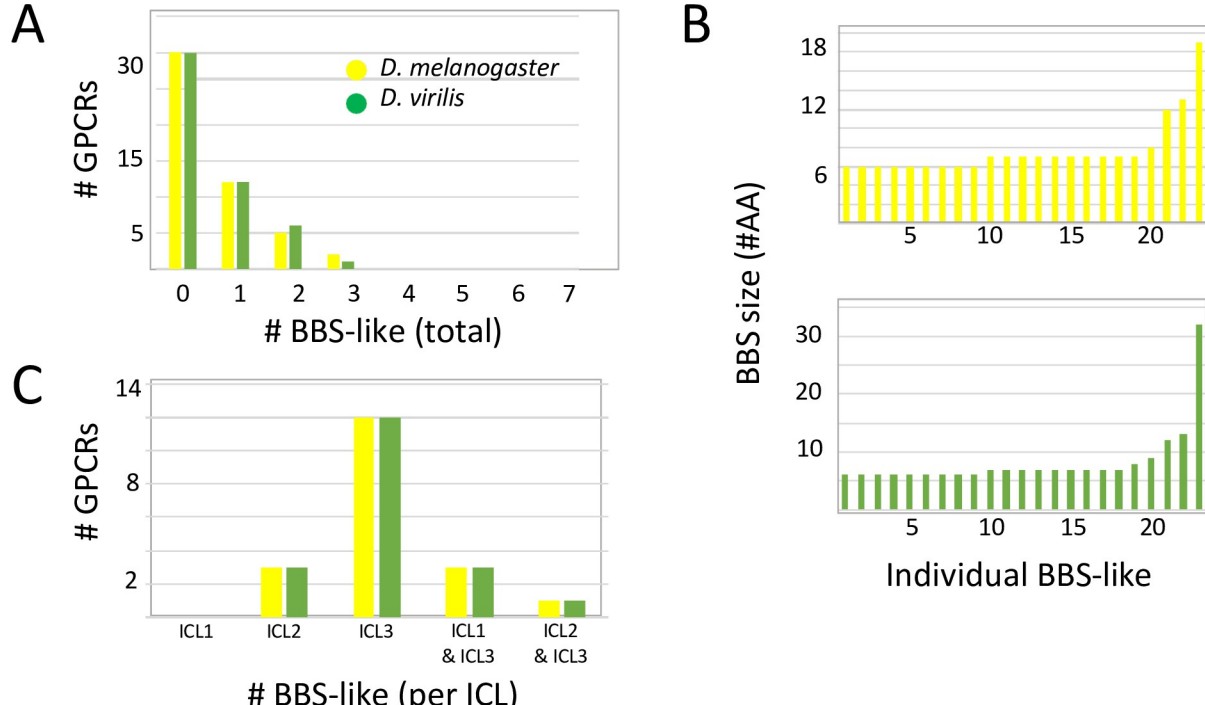

**Fig 2. A numerical overview of the incidence of conserved BBS-like sequences in the ICL domains of 49 *Drosophila* neuropeptide GPCRs.** (A). A plot of the number of BBS-like sequences per GPCR. The equal numbers across the range between *D. melanogaster* and *D. virilis* is explained by the applied definition, based on conservation between these two species. (B). The distribution of BBS-like sequence size across all occurrences for *D. melanogaster* (yellow) and *D. virilis* (green) genes. The majority correspond to the canonical 6–7 AA size, although a sizable number exhibit greater length. (C). The distribution of BBS-like sequences between the ICL domains 1, 2 and 3.

on compilations of *Drosophila* GPCR sequences which are found in the file, S1 Text. This single document collates the predicted sequences for each GPCR family, with annotations for *D. melanogaster* and for *D. virilis*. S1 Text also indicates the positions and extents of the predicted seven transmembrane (TM) domains for each receptor. Each entry lists its Genbank reference number, or alternatively, cites a Supporting Text file that documents my direct inspection of *D. virilis* genomic DNA. In many cases, such inspections produced reasonable predictions for orthologous GPCR isoforms in *D. virilis*, which were not recovered by blastp searches. Family A (rhodopsin-like) GPCRs are shown first (in Figs 3–9), and Family B (secretin receptor-like) GPCRs follow (in Figs 10–11); the presentation order is otherwise arbitrary. For each GPCR in these Figures, block diagrams illustrate the 7 TMs, the intervening loops and the CT domains for both the *D. melanogaster* (yellow) and *D. virilis* (green) representative. All sequence matches with the canonical BBS [1] that are located within intracellular domains are indicated at their approximate position. BBS's are red and above the line, if conserved across *D. melanogaster* and *D. virilis*. BBS's are black and below the line, if not conserved.

The initial set of rhodopsin-like neuropeptide GPCRs is diagrammed in Fig 3. The AstA-R1 displays a single conserved BBS near its C terminus. The paralogue GPCR, AstA-R2 does not contain any BBS candidates in its PA isoform, whereas the PB isoform (generated by stop suppression) does contain one. The BBS in the *D. virilis* AstA-R2 PB does not maintain sequence conservation, but I rated it conserved based on its precisely correspondent position within the final alternative exon. The two CCK-R–like paralogs (17D1 and 17D3) each contain a clear, well-conserved BBS in the CT domain that is different between the two. In addition, the AstA-R1 and 17D1 and 17D3 CCK-L receptors each contain a conserved BBS in the ICL2

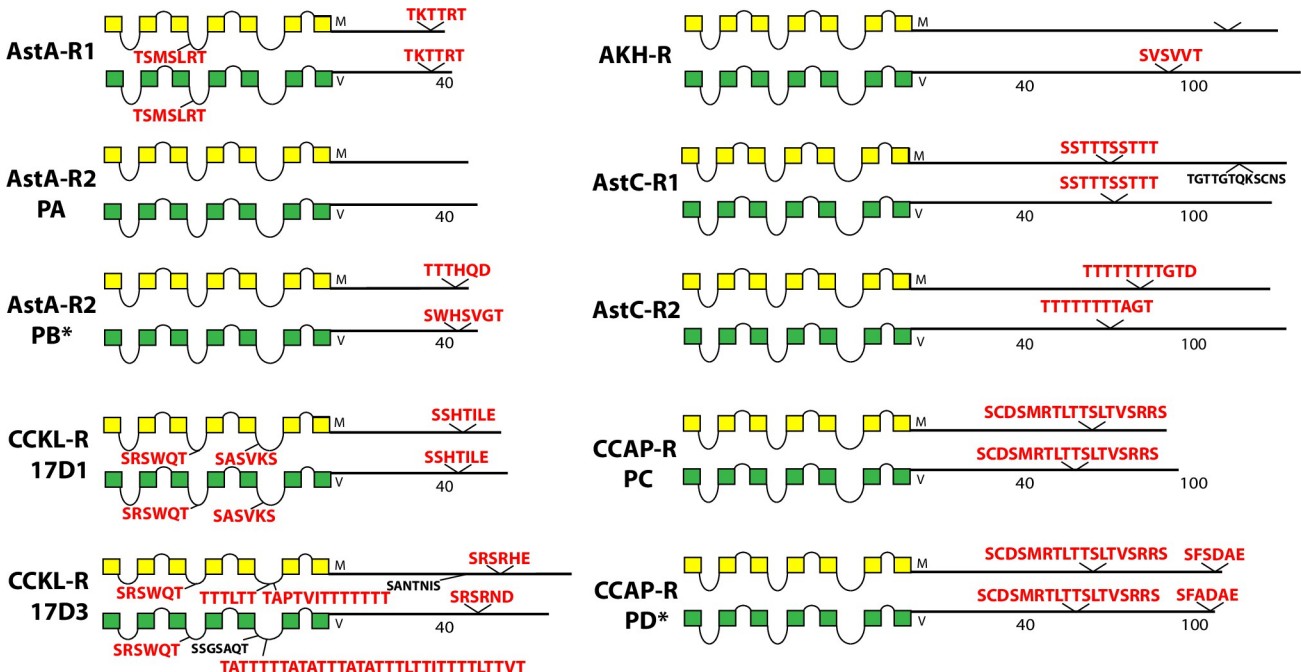

**Fig 3. Schematic representations of *D. melanogaster* (yellow) and *D. virilis* (green) examples of eight different neuropeptide GPCRs, including the AstA-R1, two isoforms of the AstA-R2, the CCKLR-17D1-R, the CCKLR-17D3-R, the AKH-R, the AstC-R1 and -R2, and two isoforms of the CCAP-R.** TM domains are represented by boxes and the lengths of the CTs are to approximate relative scale and indicated by adjoining numbers. Conserved BBS's are indicated in red typeface above the lines, while non-conserved ones are indicated in black typeface below the lines. The AstA-R2 example includes two protein isoforms that differ in the incidence of BBS's: the PB isoform of *D. virilis* (*) was not recovered by blastp search, but instead results from direct inspection of *D. virilis* genomic DNA, as documented in S2 Text. Likewise the CCAP-PD isoform in *D. virilis* (*) results from direct inspection of *D. virilis* genomic DNA, as documented in S3 Text.

(noted above), while the 17D1 and 17D3 CCK -L receptors contain two or more conserved, compound BBS-like sites in their ICL3 domains. Three other receptors in this Figure, AKH-R, and the paralogous AstC-R1 and -R2 likewise contain single, conserved BBS sequences of differing lengths. The two isoforms of the CCAP-R contain one versus two BBS 's respectively.

A second set of five receptors, FMRFa-R, sNPF-R, PK1-R1, PK2-R1 and PK2-R2, all display one or two conserved BBS's, of differing lengths and at different positions along the length of the CT (Fig 4). The sNPF-R contains a BBS-like sequence in ICL3, and the PK2-R1 and -R2 paralogues contain identical, conserved BBS-like sequences in ICL2.

Eight GPCRs including CNMa-R, SP-R, CCHa1-R and TK-R 99D, all display one or two conserved BBS's in the CT domains (Fig 5). In contrast, CCHa2-R and the NPF-R have none, while the RYa-R has three isoforms (produced by alternative splicing) in *D. melanogaster* and these display one or no BBS's. The annotation for the *D. virilis* RYa-R gene predicts only an orthologue of the PA isoform. I searched and assembled an annotation for the *D. virilis* PB form that corresponds in position and sequence features (S4 Text), but could not find a potential PC orthologue (indicated by "?"). The PA and PB isoforms in *D. virilis* contain matches with orthologous BBS's in *D. melanogaster*, although the one present in the PB isoform is a match in position only. The Tk-R 86C protein contains a single BBS-like sequence in ICL3. In *D. melanogaster*, Tk-R 86C also presents a splice isoform that inserts additional sequence into the CT which includes a BBS match (S1 Text). I searched for an orthologous isoform in *D. virilis*, but found only a "low-confidence" candidate (S5 Text) that is not presented in Fig 5.

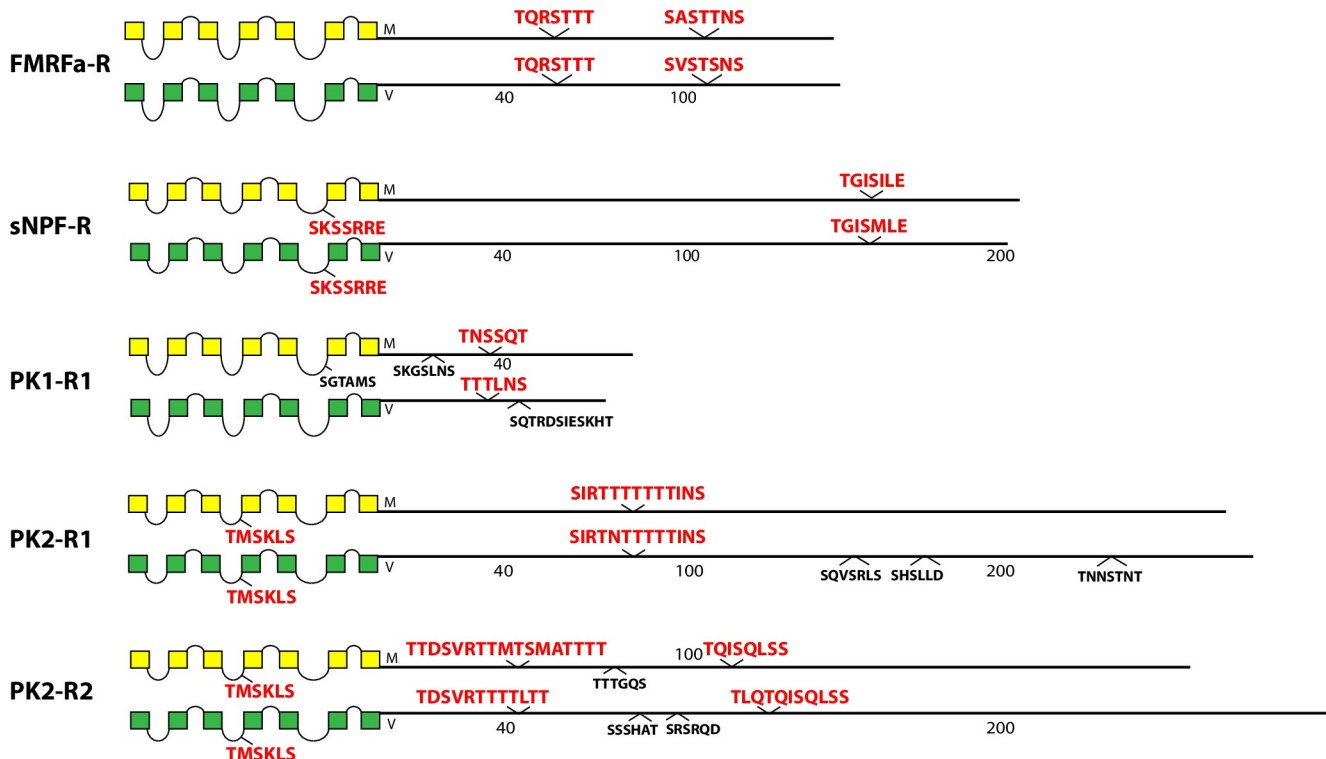

**Fig 4. Schematic representations of *D. melanogaster* (yellow) and *D. virilis* (green) examples of five different neuropeptide GPCRs, including the FMRFa-R, the sNPF-R, the PK1-R, the PK2-R1, the PK2-R2 GPCRs.**

The five receptors in Fig 6 all contain one or more conserved BBS's in their CTs. SIF-R contains seven, which is by far the largest number among all 49 *Drosophila* neuropeptide GPCRs (Fig 1A). In the aggregate, the conserved BBS sequences in the SIFa receptor account for nearly 30% of its 260 AA CT domain. The ETH receptors (isoforms PA and PB generated by alternative splicing) have completely distinct CT domains with isoform-specific BBS's. Each ETH-R BBS is highly conserved. CRZ-R also contains a conserved BBS-like sequence in ICL3.

Five of the eight GPCRs in Fig 7 do not contain a conserved BBS in the CT domains–the paralogous MS-R1 and -R2, LGR1, Proc-R and the CAPA-R. Whereas the Trissin-R isoforms, and the two other large glycoprotein receptors LGR3 and LGR4 all contain one or two precisely-conserved BBS's. Several receptors display BBS-like sequences in ICL domains, some in multiple ICLs (LGR4) and others with as many as three distinct BBS-like sequences in a single ICL (Trissin-R). In *D. melanogaster*, the CAPA-R presents an alternatively spliced isoform in which the receptor is truncated near the 6[th] TM domain. I searched for a similar situation in the *D. virilis* genomic DNA, but found no compelling evidence (S6 Text), so the existence of that PC isoform in *D. virilis* (not represented) remains uncertain. In *D. melanogaster*, Trissin-R exhibits 3 different isoforms by alternative splicing. For *D. virilis*, only the PB isoform is predicted; I found evidence for orthologues of the PC and PD isoforms from direct inspection of genomic DNA (S7 Text). Notably, isoform PD differs from PB and PC by deleting the second BBS of the CT. In addition, alternative splicing generates the Trissin-R PC isoform which is distinguished by an altered BBS-like sequence in ICL3. The sequences of all BBS-like sites in the three Trissin-R isoforms match the BBS model, but the sequence in the PC isoform now contains three phosphorylatable resides, instead of just two and its adjacent sequences are different (S3 Fig).

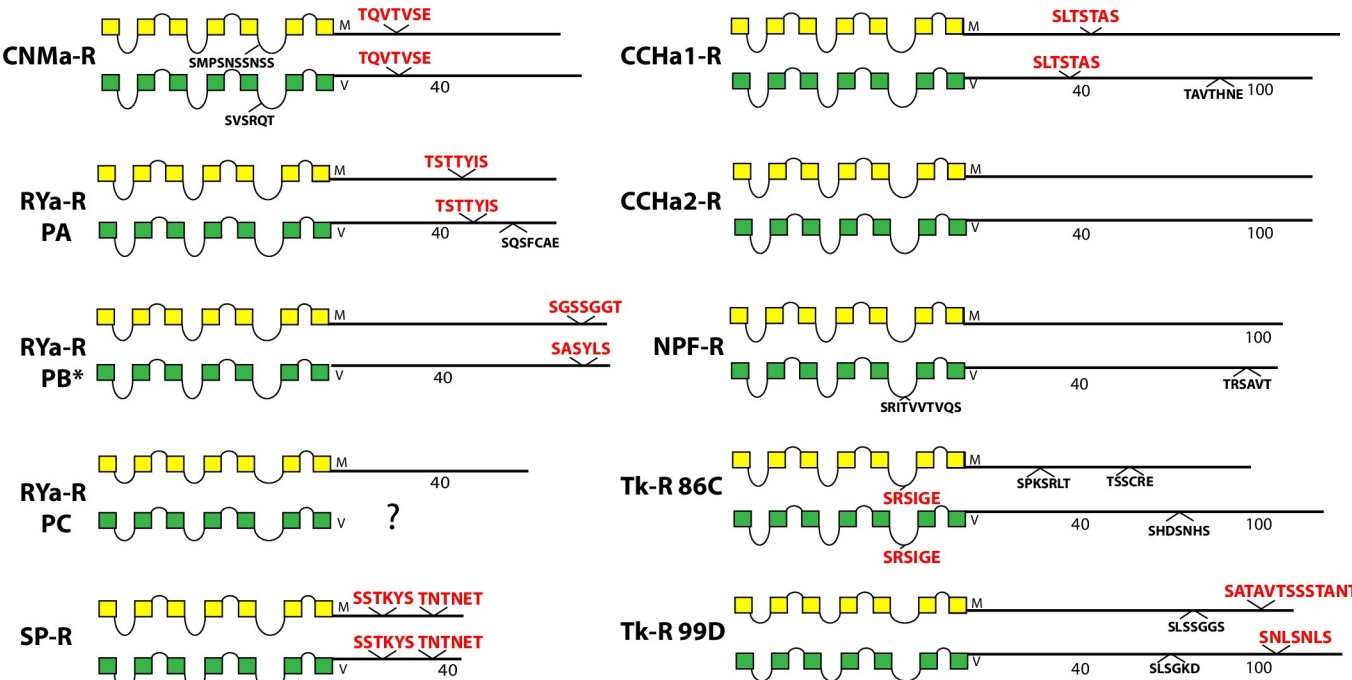

**Fig 5. Schematic representations of *D. melanogaster* (yellow) and *D. virilis* (green) examples of eight different neuropeptide GPCRs, including the CNMa-R, the RYa-R, the SP-R, the CCHa1-R, the CCHa2-R, the NPF-R, the Tk-R 86C, and the Tk-R 99D.** The RYa-R example includes three different predicted *D. melanogaster* isoforms; for *D. virilis*, the PA form is predicted, while evidence for the existence of PB(*) derives from direct inspection of genomic DNA (S4 Text) and no evidence for PC was forthcoming. PC therefore remains uncertain in this species, as indicated by the question mark.

Rickets, moody and the orphan receptor encoded by the gene *CG32547* all possess long CTs (300+ to 500+ AAs) (Fig 8). All contain two to four conserved BBS's, with the exception of the moody PC isoform which contained none. The moody PC was not retrieved from *D. virilis* by blast search; it resulted from direct inspection of genomic DNA records (S8 Text). Both species use alternative reading frames of the same exon to encode the differing CT isoforms. The CG32547 receptor contains a conserved BBS-like sequence in ICL3.

The final set of rhodopsin-like neuropeptide GPCRs are orphan receptors encoded by the genes *CG13575, CG12290, CG30340, CG33639, Tre1, CG13229, CG13995* and *CG4313* (Fig 9). The CG13575, CG12290 and CG13229 GPCRs contain conserved BBS's but the other five do not. Four of these GPCRs display conserved BBS-like sequences in ICL1 and/ or ICL3 domains as shown.

Among the five Family B *Drosophila* neuropeptide GPCRs (those related to the ancestor of the mammalian secretin R), the two closely-related to calcitonin R (Dh31-R and the orphan GPCR called hector, encoded by *CG4395*) display different profiles of conserved BBS sequences (Fig 10). Dh31-R contains conserved BBS's on both of its protein isoforms, whereas Hector contains none. PDF-R also presents a difference in CT structure via alternative splicing: the PA isoform does not display conserved BBS sequences, whereas the PD isoform introduces a different final ~50 AAs that includes multiple compound BBS's. PDF-R displays a single conserved BBS-like sequence in its ICL3.

Finally, the Family B GPCRs related to Corticotropin Releasing Factor Receptors, Dh44 -R1 and -R2 display one or two conserved BBS (Fig 11). A PB isoform of Dh44-R2 lacks any such. All GPCRs in this figure contain a BBS-like sequence in ICL3 (similar to that in PDF- R (Fig 10).

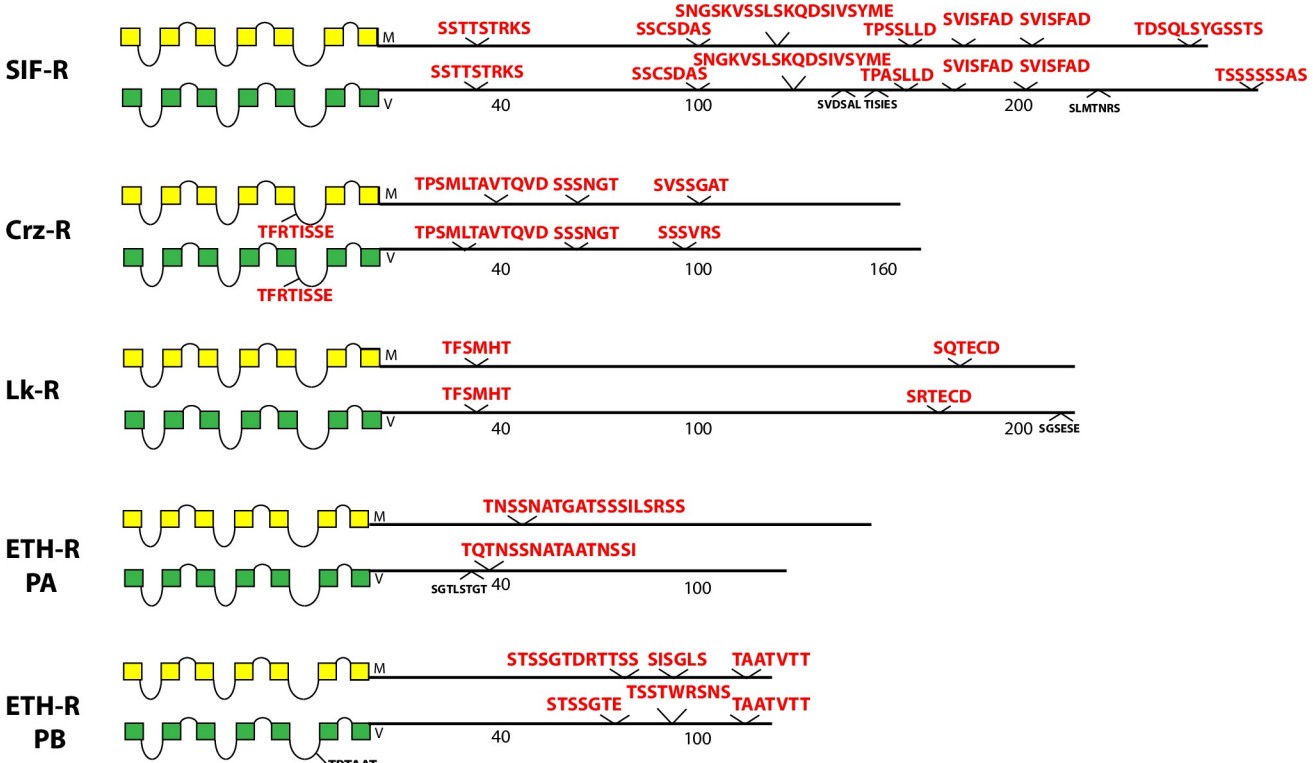

**Fig 6. Schematic representations of *D. melanogaster* (yellow) and *D. virilis* (green) examples of four different neuropeptide GPCRs, including the SIF-R, the Crz-R, the LK-R, and the ETH-R.** The ETH-R presents two different CT isoforms based on alternative splicing.

## Evaluating the utility of comparing *D. melanogaster* and *D. virilis* neuropeptide GPCRs

The comparison between the *D. melanogaster* and *D. virilis* species proved valuable to help focus this analysis on BBS sequences in the neuropeptide GPCR CTs that are the most durable across a considerable length of evolutionary time. The utility in that method is to potentially highlight the sites most likely to subserve significant physiological regulation. However, it is also possible that a comparison between them introduces false negative calls by excluding consideration of sites in *D. melanogaster* GPCR CTs that are functionally conserved in species more recently-diverged. I therefore re-examined the ten GPCRs that failed to display conserved CT BBS's in the initial screen, to ask whether more limited BBS sequence conservation might be revealed. For each of these GPCRs, I considered 15 additional species from the *Sophophora* sub-genus (which includes *D. melanogaster*) and two additional species from the *Drosophila* sub-group (which includes *D. virilis*). Documentation of sequence predictions and alignments for each of the ten GPCRs are found in S9–S18 Texts. The results were mixed and below I sort outcomes to three different categories. In the accompanying figures (Figs 12–23), precise BBS matches that display some conservation across species are marked in red and above the line (as in previous figures). Also and as before, BBS's are black and below the line, if not conserved. However, these figures add a third feature: the sequence of BBS-correspondent positions in those species that do not match the BBS model: these are black, below the line and in italics. I detail these to allow direct consideration of (for example) the possible involvement of partial BBS codes [1]. In sum, the results warrant some re-consideration of BBS conservation in neuropeptide GPCRs across the *Drosophila* genus.

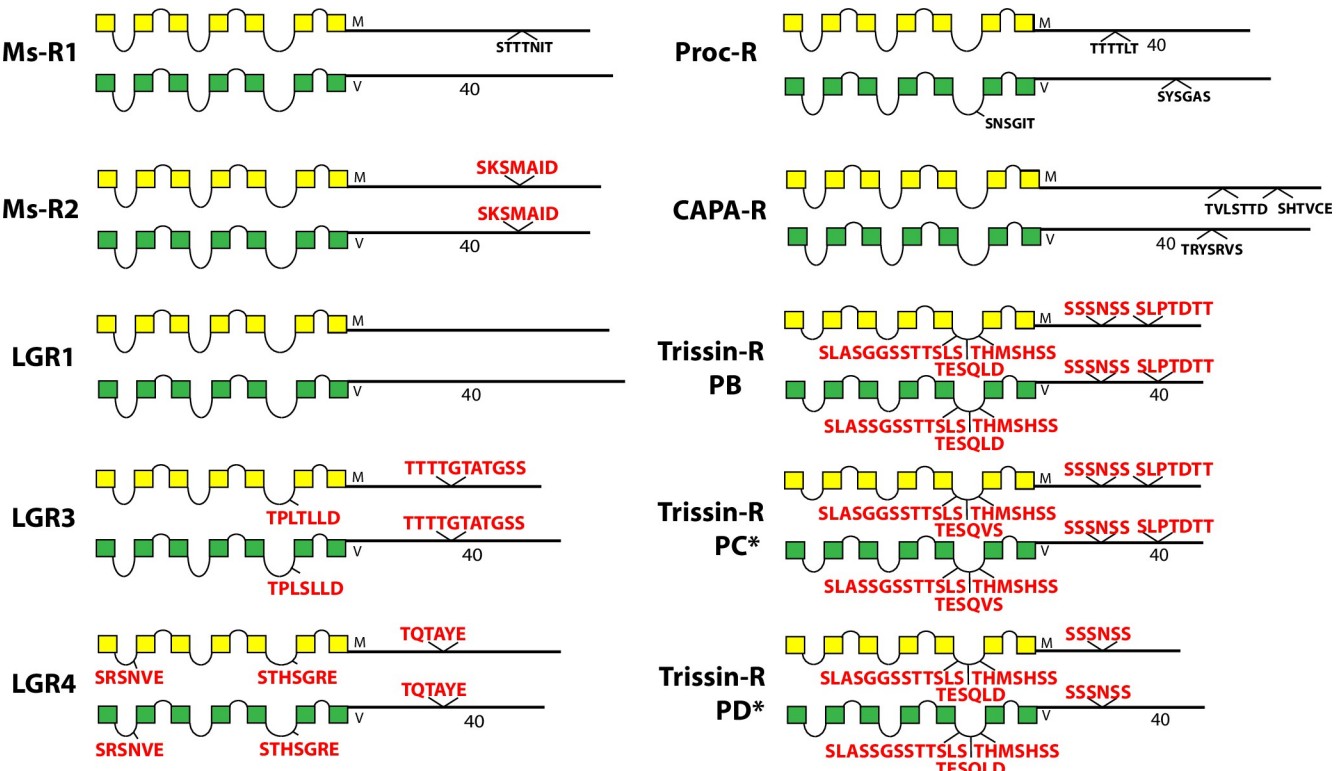

**Fig 7. Schematic representations of *D. melanogaster* (yellow) and *D. virilis* (green) examples of eight different neuropeptide GPCRs, including the Ms-R1, Ms-R2, LGR1, LGR3 and LGR4, Proc-R, the CAPA-R and Trissin-R.** With the exception of the Trissin-R group, the others contain one or no conserved BBS's. The PC and PD isoforms of *D. virilis* (*) were not recovered by blastp search, but instead results from direct inspection of *D. virilis* genomic DNA, as documented in S7 Text. The Trissin-R group displays one or two conserved BBS's and three conserved BBS-like sequences in the 3rd ICL. The second BBS-like sequence (TESQLD) is modulated by alternative splicing in isoform PC (cf. S1 Fig).

**Category 1. GPCRs displaying broad BBS sequence conservation across the *Sophophora* sub-genus.** Among the 10 *D. melanogaster* GPCRs re-examined that lacked any conserved BBS (conserved as far as to *D. virilis*), four had CT BBS matches that extended clearly across most or all of the 15 species selected from the *Sophophora* sub-genus. These four included the Proc-R (Fig 12), the Tk-R86C (Fig 13), the CAPA-R (Fig 14) and the PA isoform of the PDF R (Fig 15). In *D. melanogaster*, the Proc-R contains a single match with the BBS prediction (TTTTLT), but it was not sustained in *D. virilis* nor the other representatives of the *Drosophila* sub-genus, due to the change from 'T' at the final position to an 'N'. However, in each of the 15 additional *Sophophora* species examined, the BBS was perfectly conserved, which strongly suggests it indeed plays a functional role. Likewise, the two BBS matches in the Tk-R 86C CT of *melanogaster* were not conserved in *D. virilis*, but were conserved throughout the *Sophophora* (with one exception). In the CAPA-R, a similar situation is seen wherein one or both of two BBS matches extends widely across the *Sophophora* species, but not as far as *D. virilis*. Lastly, the PA isoform of PDF-R contains a BBS near the 7th TM which is not conserved in *D. virilis*, but is found in all other *Sophophora* flies examined. These examples illustrate that, in certain GPCRs, the lack of evolutionary conservation of CT BBS's between *D. melanogaster* and *D. virilis* may represent false negatives in this analysis.

**Category 2. GPCRs displaying very limited BBS conservation across *Sophophora*.** Two of the 10 GPCRs were difficult to evaluate in that, certain BBS's displayed clear conservation but which was limited to only two or three of the additional 15 species examined, at most.

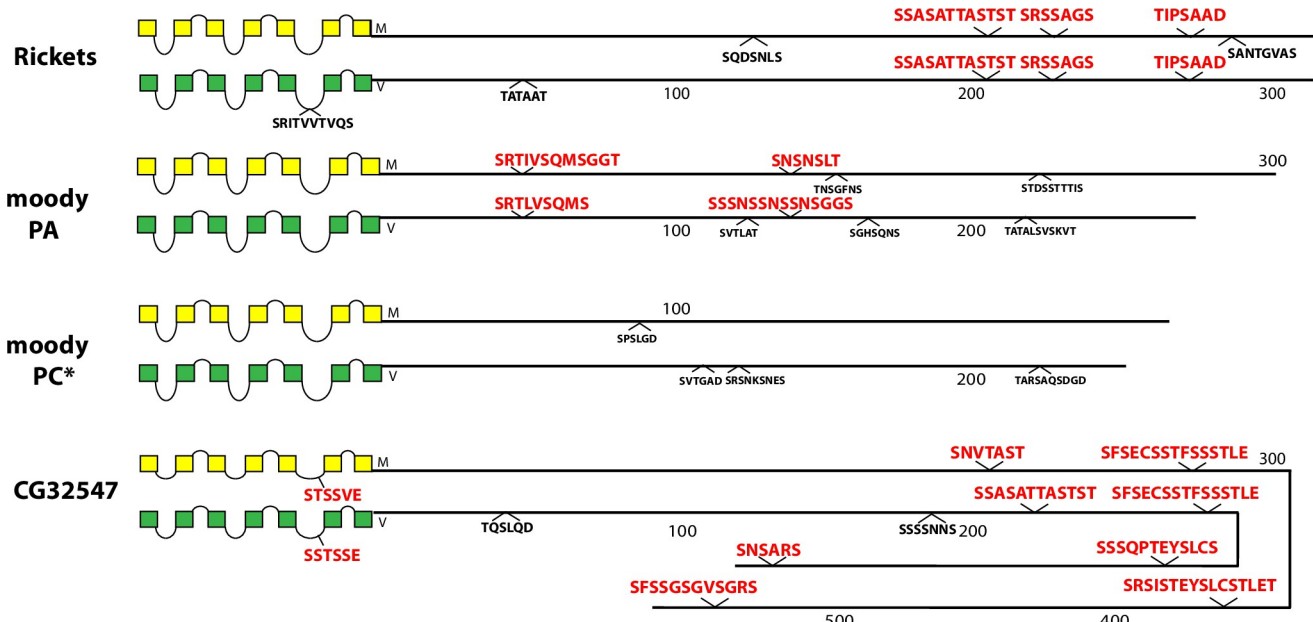

**Fig 8. Schematic representations of *D. melanogaster* (yellow) and *D. virilis* (green) examples of three different neuropeptide GPCRs, that display extensive CT length, including the Rk, moody, the LK-R, and the orphan GPCR encoded by *CG32547*.** All three exhibit a series of conserved BBS's. The *moody* locus encodes two isoforms that present very different BBS profiles; the PC isoform in *D. virilis* (*) was not recovered by a blastp search but derives from direct inspection of genomic sequences (S8 Text). The final two BBS's in CG32547 are positioned correctly, but, because of space constraints, their sequences are inverted relative to the apparent direction of the CT.

These included the CCHa2-R which featured a conserved BBS near the very end of the CT, but present only in the species *D. biarmipes* and *D. suzukii* (Fig 16). Likewise the LGR1 featured a BBS conserved between *D. rhopaloa* and *D. eugracilis*, but it was not found in any other species in this analysis (Fig 17). The argument that these conserved BBS's represent functional sites for arrestin interactions is therefore much less convincing than for the ones in Category 1.

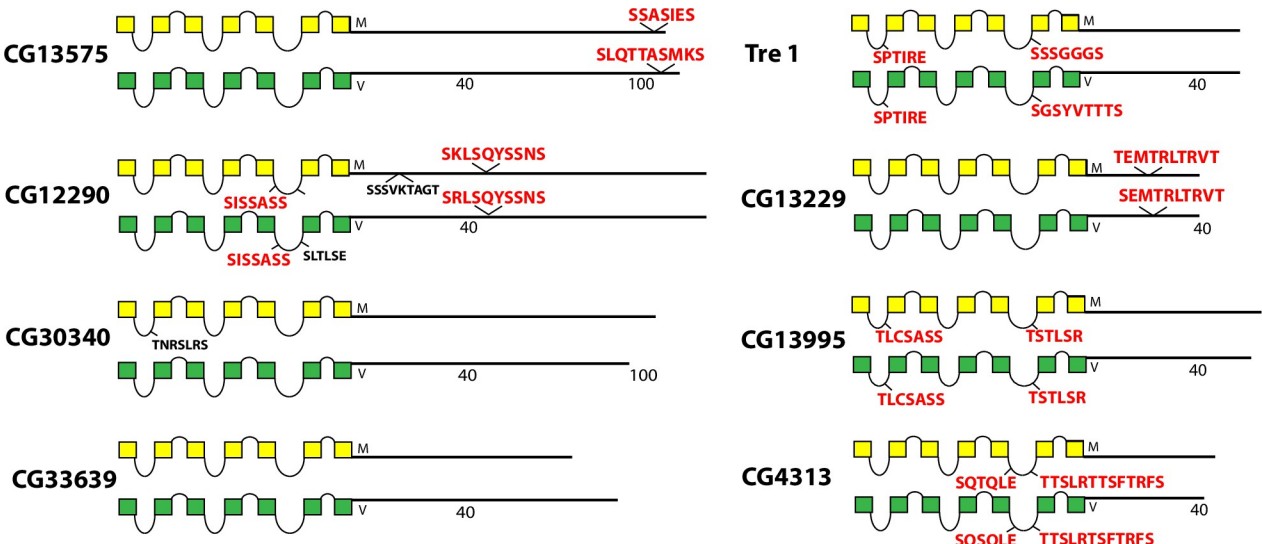

**Fig 9. Schematic representations of *D. melanogaster* (yellow) and *D. virilis* (green) examples of eight different orphan neuropeptide GPCRs, including GPCRs encoded by the *CG13575, CG12290, CG30340, CG33639*, the *Tre1, CG13229, CG13995* and *CG4313* genes.** This group exhibits a diversity of BBS profiles, as well as several receptors that display BBS-like sequences.

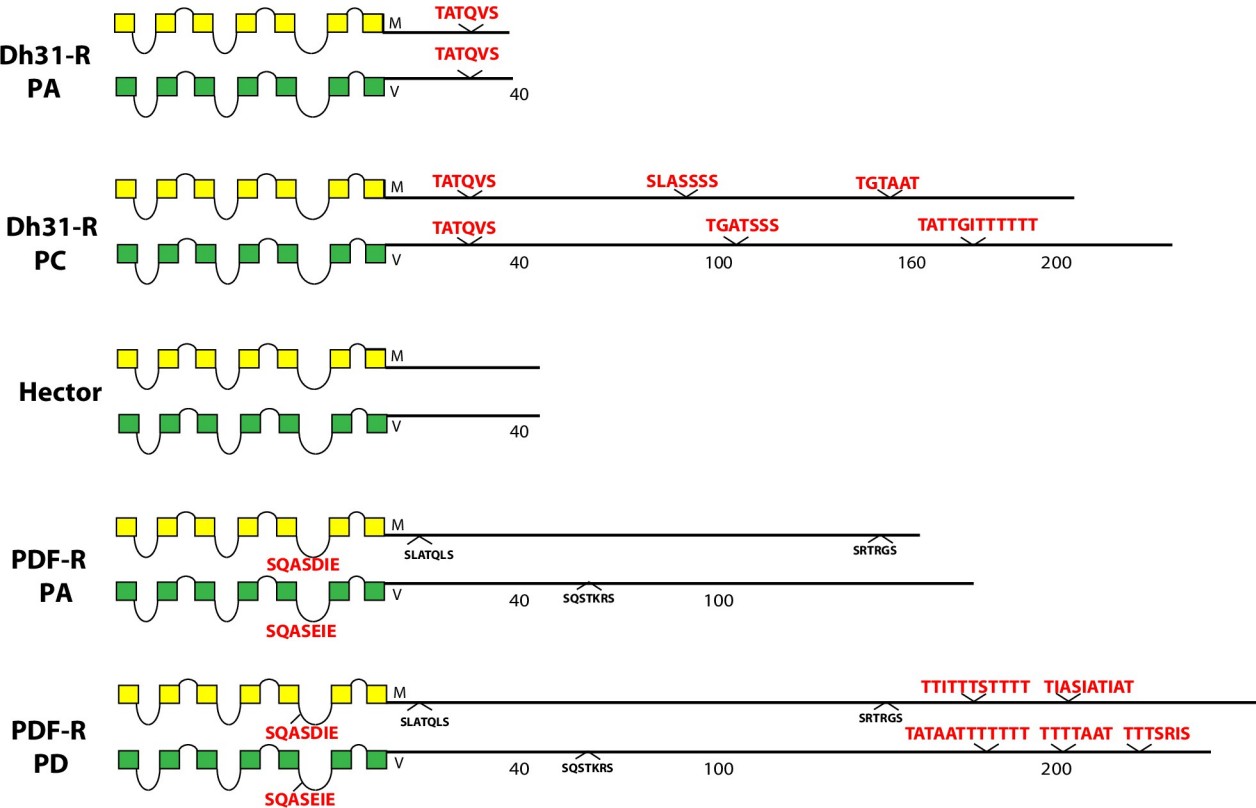

**Fig 10. Schematic representations of *D. melanogaster* (yellow) and *D. virilis* (green) examples of three different Family B neuropeptide GPCRs, including the Dh31-R, the Hector-R and PDF-R.** The Dh31-R and PDF-R presents two different CT isoforms based on alternative splicing.

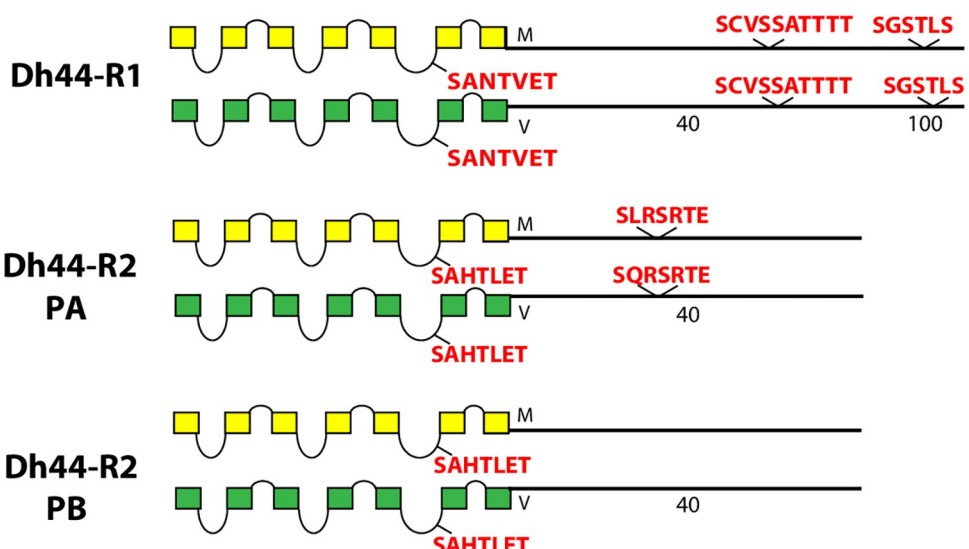

**Fig 11. Schematic representations of *D. melanogaster* (yellow) and *D. virilis* (green) examples of two different Family B neuropeptide GPCRs, including the Dh44-R1 and Dh44-R2.** The Dh44-R2 presents two different CT isoforms based on alternative splicing.

# Proc-R

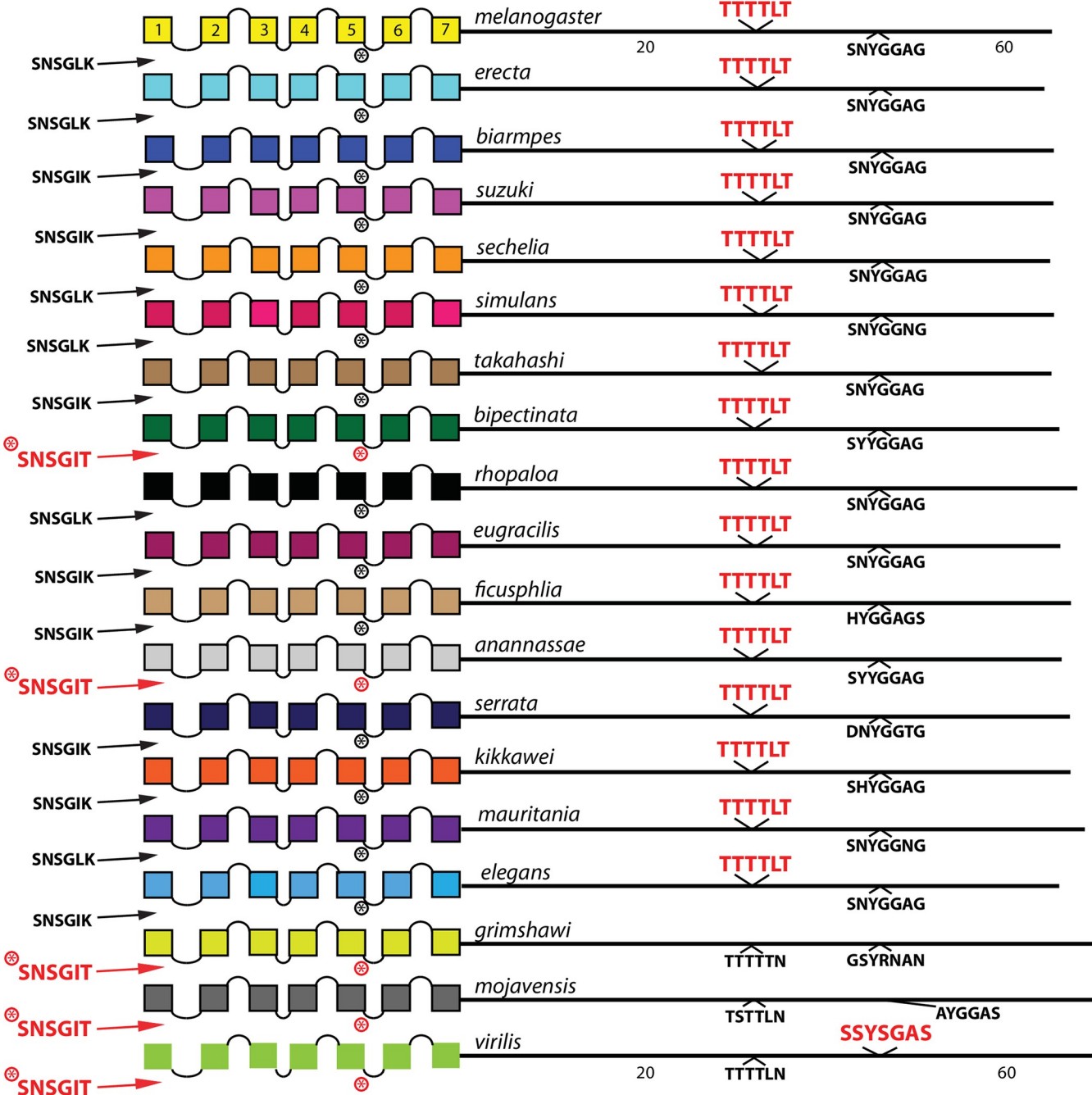

**Fig 12. Extended evolutionary comparison of the Proctolin receptor.** Schematic representations from 19 different *Drosophila* species, including *D. melanogaster* (yellow, top) and *virilis* (green, bottom). The final three (*D. grimshawi*, *D. mojavensis* and *D. virilis*) are all members of the sub-genus *Drosophila*; the other 16 are members of the sub-genus *Sophophora*. Whereas a comparison limited to just *D. melanogaster* and *D. virilis* indicated no conserved BBS sequences in the Proc-R (Fig 7), the expanded roster of species shows extensive BBS conservation across species more closely related to *D. melanogaster*. As before, conserved BBS sequences and positions are indicated by red lettering; non-conserved ones are indicated in black.

# Tk-R 86C

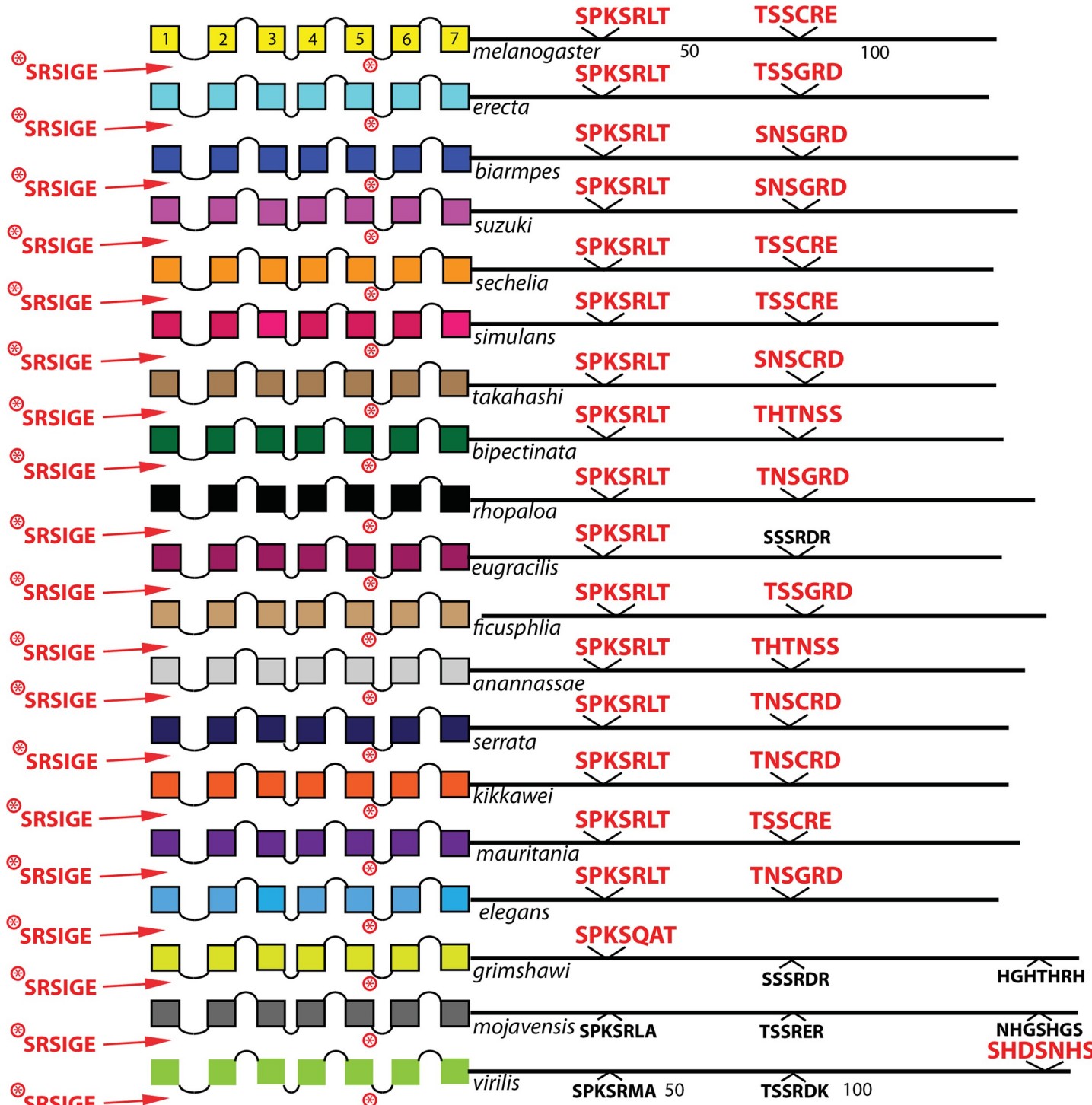

**Fig 13. Extended evolutionary comparison of the Tk-R 86C.** Schematic representations from 19 different *Drosophila* species, including *D. melanogaster* (yellow, top) and *D. virilis* (green, bottom). See Fig 12 Legend for more detail. Whereas a comparison limited to just *D. melanogaster* and *D. virilis* indicated no conserved BBS sequences in the Tk-R 86C R (Fig 5), the expanded roster of species shows extensive BBS conservation across all *Sophophora* species.

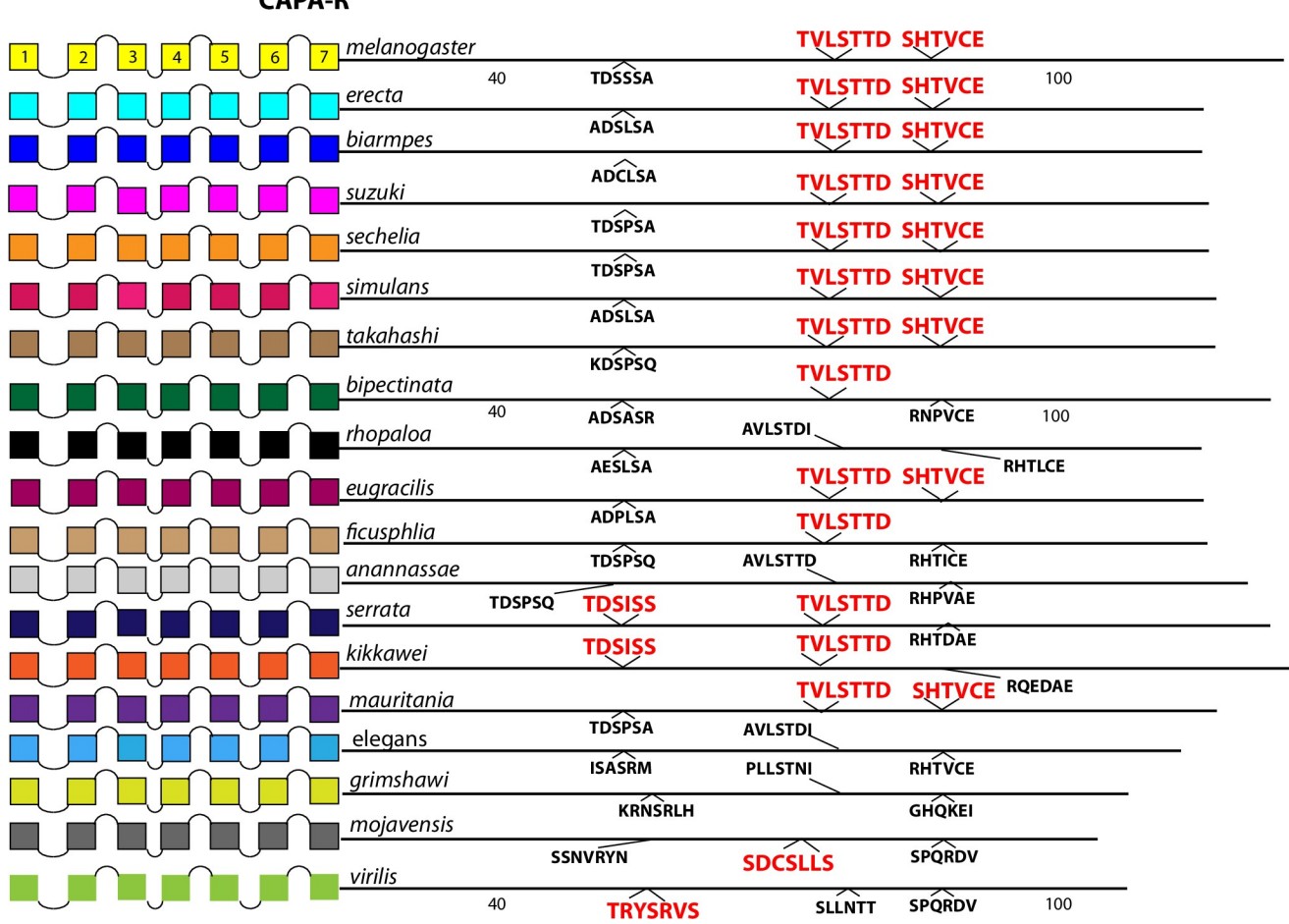

**Fig 14. Extended evolutionary comparison of the CAPA-R.** Schematic representations from 19 different *Drosophila* species, including *D. melanogaster* (yellow, top) and *D. virilis* (green, bottom). See Fig 12 Legend for more detail. Whereas a comparison limited to just *D. melanogaster* and *D. virilis* indicated no conserved BBS sequences in the CAPA-R (Fig 7), the expanded roster of species shows extensive BBS conservation across most *Sophophora* species. Likewise the BBS-like sequence in ICL3 is conserved across three species of the sub-genus *Drosophila*.

**Category 3. GPCRs displaying no BBS conservation across *Sophophora*.** Four GPCRs failed to reveal any conservation of BBS sequences across all *Sophophora*. These included the NPF-R (Fig 18), the Family B receptor named Hector (Fig 19), and the orphan receptors encoded by *CG33639* (Fig 20) and by *CG30340* (Fig 21). The single BBS match found in the NPF-R CT of *D. virilis* ("TRSAVT", Fig 5) was conserved in the other two species of the *Drosophila* sub-genus (Fig 18), but the corresponding position in all *Sophophora* species ("LRSAIT") failed to match the BBS prediction. This sequence within the NPF-R CT was not unique in displaying constancy across *Sophophora* species, and so I judge it is not a cryptic or degenerate BBS, although it would classify as a partial BBS, according to Zhou *et al*. [1]. The CTs of the Hector (Fig 19) and the CG33639 GPCRs (Fig 20) were devoid of candidate BBS's in all species examined. Finally, the orphan receptor encoded by *CG30340* displayed several candidate BBS's among the various species examined, but no sites were clearly conserved among the *Sophophora* species. One site did reveal evolutionary conservation between *D. anannassae* and two of the *Drosophila* sub-genus groups, *D. grimshawi* and *D. mojavensis* (Fig 21). In sum, this third category of receptors supports the contention that certain *Drosophila*

**PDFR-PA**

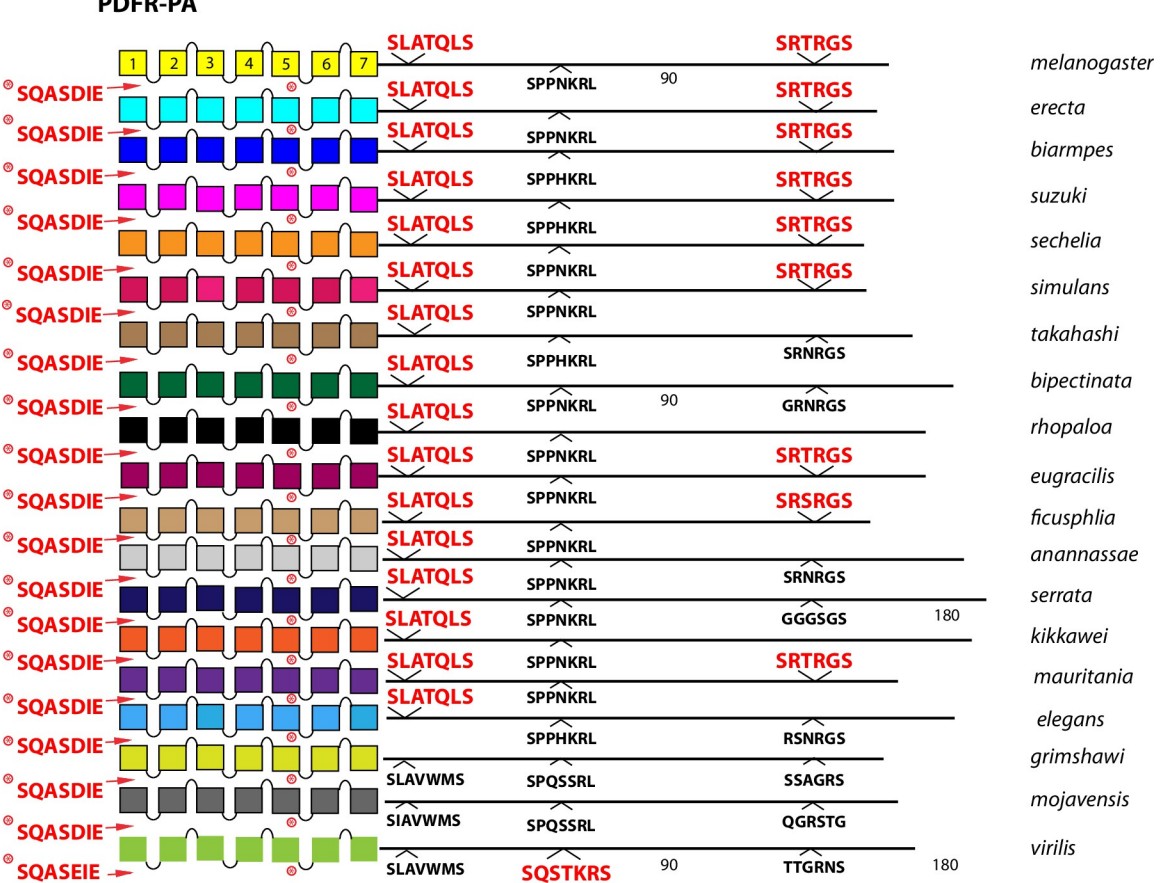

**Fig 15. Extended evolutionary comparison of the PA isoform of PDF-R.** Schematic representations from 19 different *Drosophila* species, including *D. melanogaster* (yellow, top) and *D. virilis* (green, bottom). See Fig 12 Legend for more detail. Whereas a comparison limited to just *D. melanogaster* and *D. virilis* indicated no conserved BBS sequences in the PDF-R PA isoform (Fig 10), the expanded roster of species shows extensive BBS conservation soon after the 7th TM across all *Sophophora* species and conservation of a 2nd BBS near the end of the CT across many.

neuropeptide GPCRs lack any conserved CT BBS's, as predicted from the initial comparison between *D. melanogaster* and *D. virilis* sequences.

## Evaluating evolutionary consistency in isoform-specific BBS content

21 *Drosophila* neuropeptide GPCRs are expressed as a set of protein isoforms with alternative C termini (due to alternative splicing or to Stop suppression). Among these, eight present isoforms with differences in the profile of conserved BBS's. AstA-R2 (Fig 3), RYa-R (Fig 5), ETH-R (Fig 6), Trissin-R (Fig 7), moody (Fig 8), DH31-R and PDF-R (Fig 10), and DH44-R2 (Fig 11). I therefore asked whether the *D. melanogaster*–to—*D. virilis* comparison, which indicated conservation in many examples, provided a valid prediction across evolution for such differences between GPCR isoforms. As examples, I considered the PDF-R and AstA-R2 examples. The PDFR-PD isoform is distinguished from the PA by the notable presence of several compound BBS's near the end of the CT. As shown in Fig 22, this BBS profile is a consistent feature of the PDF R PD isoform across all species of the *Sophophora* and *Drosophila* subgenera examined, and therefore may in fact play a substantial functional role in PDFR PD signaling, trafficking and turnover. The AstA-R2 likewise displays isoform-specific conservation

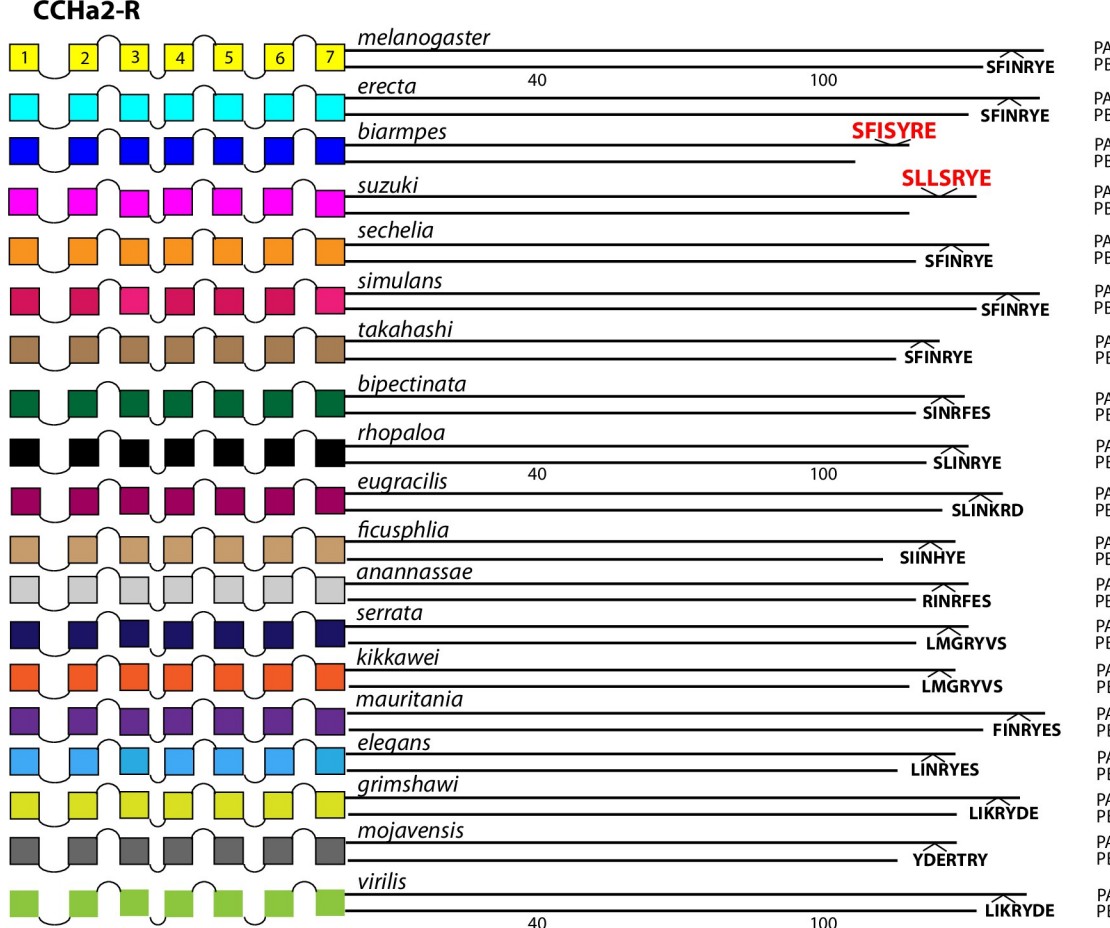

**Fig 16. Extended evolutionary comparison of the CCHa2-R.** Schematic representations from 19 different *Drosophila* species, including *D. melanogaster* (yellow, top) and *D. virilis* (green, bottom). See Fig 12 Legend for more detail. An initial comparison limited to just *D. melanogaster* and *D. virilis* indicated no conserved BBS sequences in the CCHa2-R (Fig 5), and consideration of the expanded roster of species confirms that conclusion. *D. biarmpes* and *D. suzukii* share a common BBS sequence near the end of the CT, but the cognate sequence in the other species does not support a hypothesis of extensive BBS conservation.

in its BBS profile, but as a counter-example to the PDF-R one: the lack of any BBS in the PA isoform appears to be a durable feature across all *Sophophora* species (Fig 23). These results suggest the *D. melanogaster*–to—*D. virilis* comparison produces useful predictions for the presence and absence of BBS's that vary across GPCR isoforms.

### Evaluating GPCR arrestin binding sites with results from prior *in vitro* β-arrestin:GFP assays

Upon functional expression in *hEK-293* cells, β-arrestin2:GFP is a cytoplasmic protein but can translocate to the membrane when a co-expressed GPCR is exposed to its cognate ligand [25, 39]. While some ligand:GPCR pairs do not efficiently recruit β-arrestin2:GFP, many will do so, and robust responses have been reported for numerous mammalian neuropeptide GPCRs as well as GPCRs sensitive to small transmitters like dopamine and serotonin. These studies revealed two broad categories of GPCR: β-arrestin2:GFP interactions, called A and B [40]. Category A interactions are relatively weak with the molecules dissociating while still close to the plasma membrane and prior to any GPCR internalization. In contrast Category B type

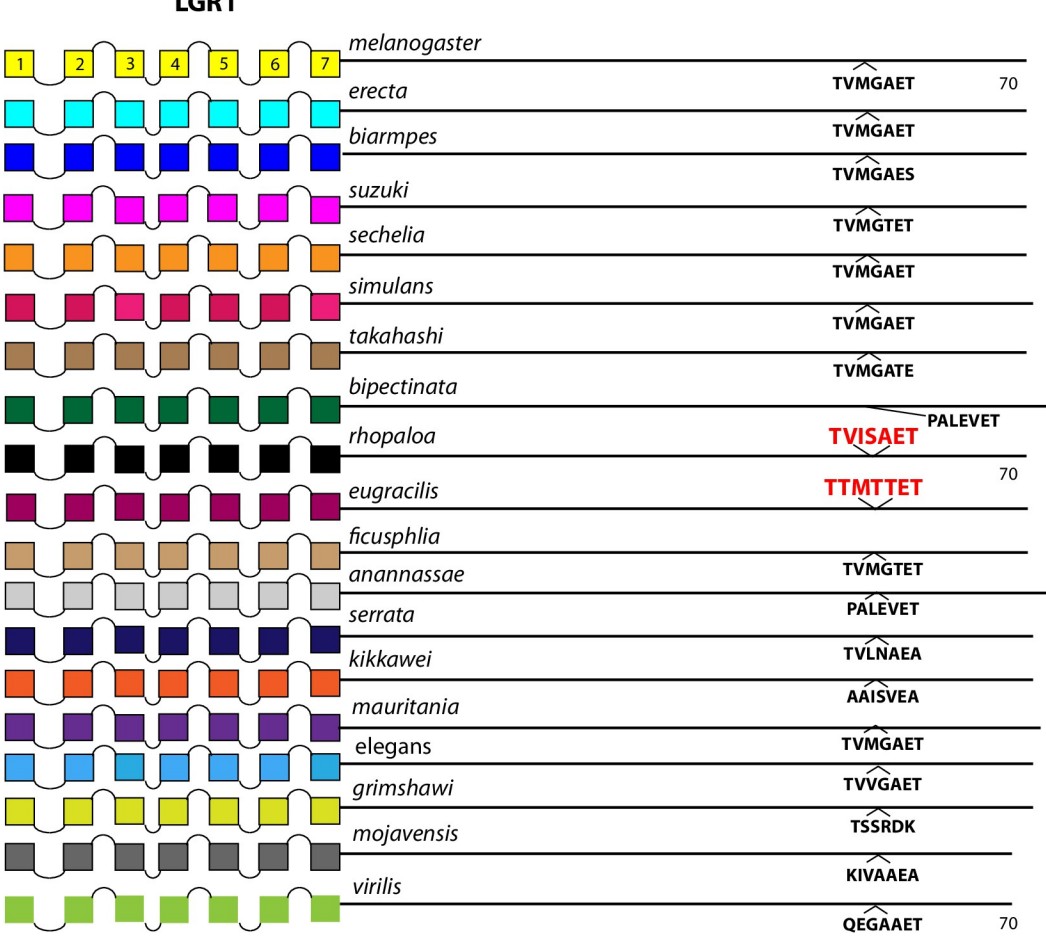

**Fig 17. Extended evolutionary comparison of the LGR1.** Schematic representations from 19 different *Drosophila* species, including *D. melanogaster* (yellow, top) and *D. virilis* (green, bottom). See Fig 12 Legend for more detail. An initial comparison limited to just *D. melanogaster* and *D. virilis* indicated no conserved BBS sequences in the LGR1 (Fig 7), and consideration of the expanded roster of species confirms that conclusion. *D. rhopaloa* and *D. eugracilis* share a common BBS sequence near the end of the CT, but the cognate sequence in the other species does not support a hypothesis of extensive BBS conservation.

interactions appear stronger, such the fluorescent β-arrestin2 remains associated with the GPCR, even as the latter is endocytosed. Oakley *et al.* [18] identified specific Ser residues in the CT of the vasopressin V2 receptor (V2R) whose phosphorylation was required for retained association with β-arrestin2 during endocytosis. This <u>Ser</u> cluster starts at the end of a complete BBS match (SCTTA<u>SSS</u>). Differential association with arrestins may have functional significance due to alternative modes of signaling that are facilitated by the scaffolding functions of β-arrestin [22].

Prior studies have shown that, when functionally expressed in *hEK-293* cells and/ or *Drosophila* S2 cells, 15 of 16 *D melanogaster* neuropeptide GPCRs tested can recruit β-arrestin2: GFP upon exposure to their cognate ligands [41–44]. Among the 16 *Drosophila* neuropeptide GPCRs tested in this paradigm, only PDFR-PA failed to recruit β-arrestin2:GFP in response to its cognate peptide ligand. 13 of these receptors were categorized with respect A versus B type interactions with β-arrestin2:GFP. The following eight receptors displayed category A-type interactions: FMRF-R, Tk-R 86C, Proc-R, AstC-R1 and AstC-R2, Ms-R1 and Ms-R2, CCHa-

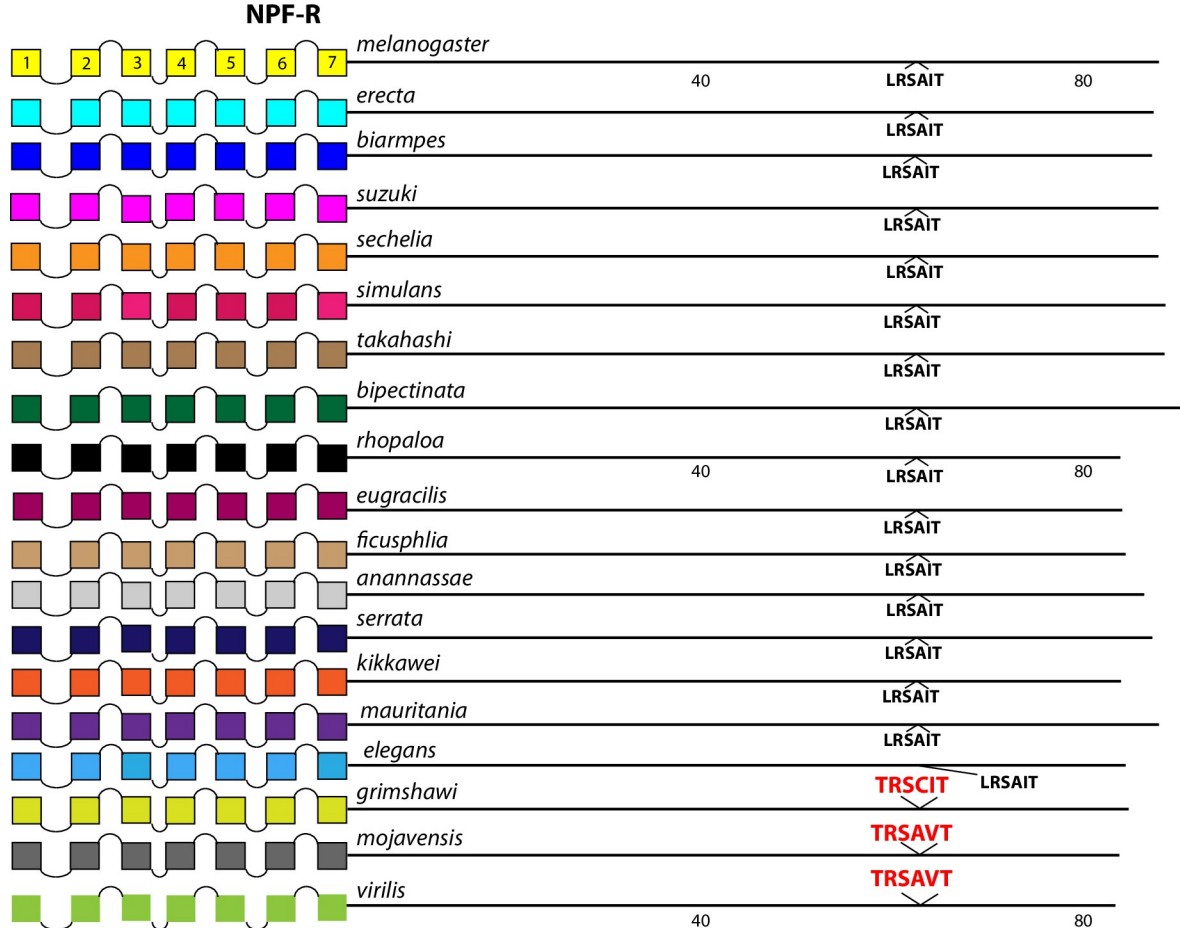

**Fig 18. Extended evolutionary comparison of the NPF-R.** Schematic representations from 19 different *Drosophila* species, including *D. melanogaster* (yellow, top) and *D. virilis* (green, bottom). See Fig 12 Legend for more detail. An initial comparison limited to just D. melanogaster and D. virilis indicated no conserved BBS sequences in the NPF-R (Fig 5), and this consideration of an expanded roster of species confirms that conclusion.

R1, LK-R and SIF-R. The following five displayed category B-type interactions: NPF-R, Crz-R, Tk-R 99D, DH44-R1 and CCAP-R. Scanning the sequences of these 13 receptors, I found no consistent BBS features (size, number or position) that correlated with apparent strength of interactions with β-arrestin2:GFP. For example, AstC-R1 and -R2 recruited β-arrestin2:GFP with a Category A pattern [42], yet both contain a single long, complete and evolutionarily-conserved BBS on the CT (albeit with different specific sequences). Likewise there was no consistent BBS sequence feature that correlated with the display of category A versus B behavior. For example, both CRZ and NPF receptors display category B behavior, yet while the Crz-R CT contains 4 separate, complete and evolutionarily-conserved BBS's, the NPF-R CT contains none.

## Discussion

The de-sensitization, endocytosis, trafficking and possible recycling of numerous GPCRs are triggered by receptor phosphorylation; the underlying mechanisms have been subjects of intense study [5, 21, 45, 46]. Keys to that evaluation are the identification of phosphorylated GPCR residues, and sequence manipulation to assess their contributions to GPCR regulation.

**Hector-R**

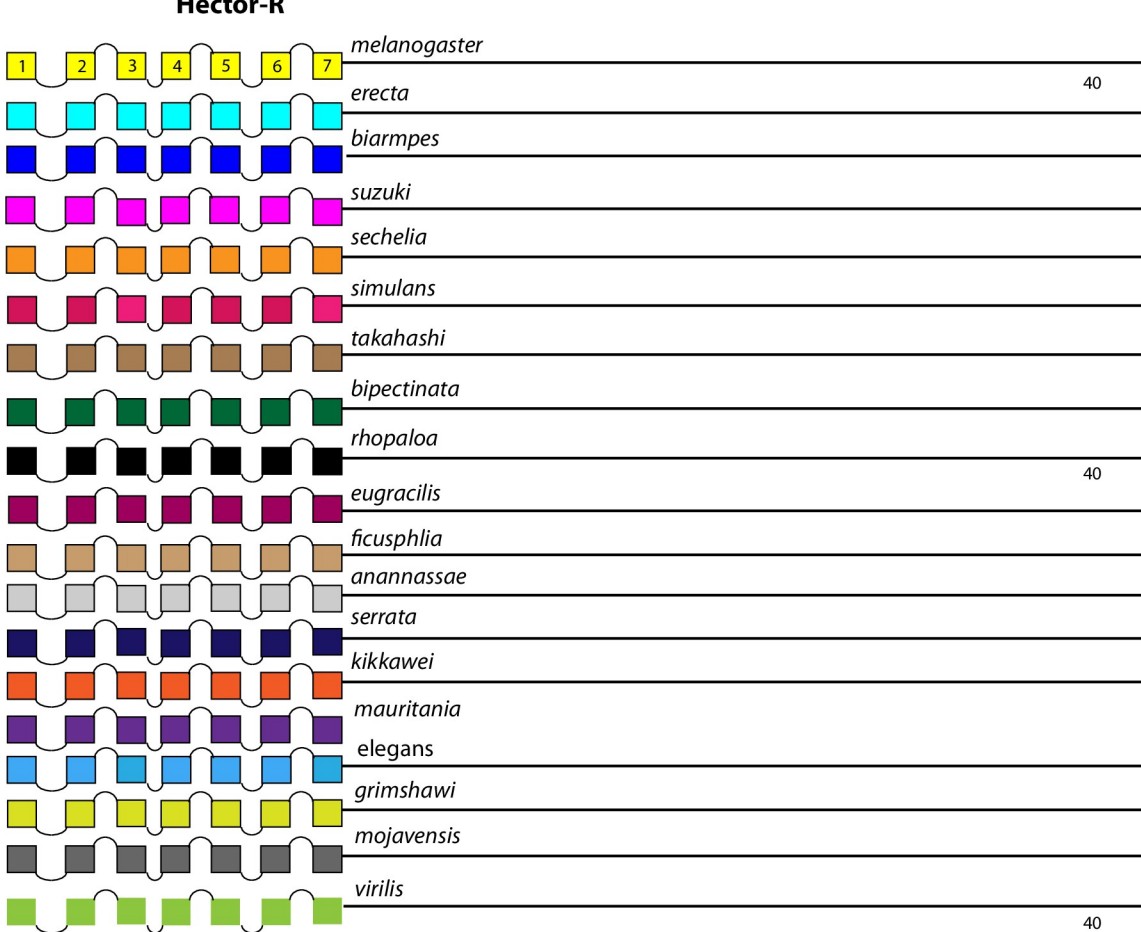

**Fig 19. Extended evolutionary comparison of the Family B1 Hector GPCR.** Schematic representations from 19 different *Drosophila* species, including *D. melanogaster* (yellow, top) and *D. virilis* (green, bottom). See Fig 12 Legend for more detail. An initial comparison limited to just *D. melanogaster* and *D. virilis* indicated no conserved BBS sequences in the Hector receptor (Fig 10), and this consideration of an expanded roster of species confirms that conclusion.

Two broad questions frame much of this work. First, how does phosphorylation trigger the subsequent steps? Do the critical modifications represent phosphorylation of a specific subset of GPCR residues, or do they instead reflect an increase in the aggregate amount of phosphorylation that requires no specific pattern. There is evidence to support both possible mechanisms, for example as deduced from studies of melanopsin [OPN4–47, 48] and of neuropeptide PDF receptor [34]. A second major question asks, can phosphorylation of distinct residues on a single GPCR direct different outcomes for signaling and desensitization? There is increasing evidence for this hypothesis, whereby different ligands engender distinct conformational states from a single GPCR thereby recruiting phosphorylation of specific but alternative residues by different kinases [40, 49]. Such "biased agonism" can direct alternative signaling pathways that display either canonical G protein-dependence, or instead β-arrestin-dependence [22]. The proposal for a consensus BBS sequence that underlies high affinity binding of arrestins to GPCRs by Zhou *et al.* [1], introduced a potentially unifying concept in this general field. That concept was the original impetus propelling the work reported herein to assess BBS prevalence among the ~50 neuropeptide GPCRs in the fly genome, and it may allow consideration of several inter-related questions. For example, the simplest asks: do

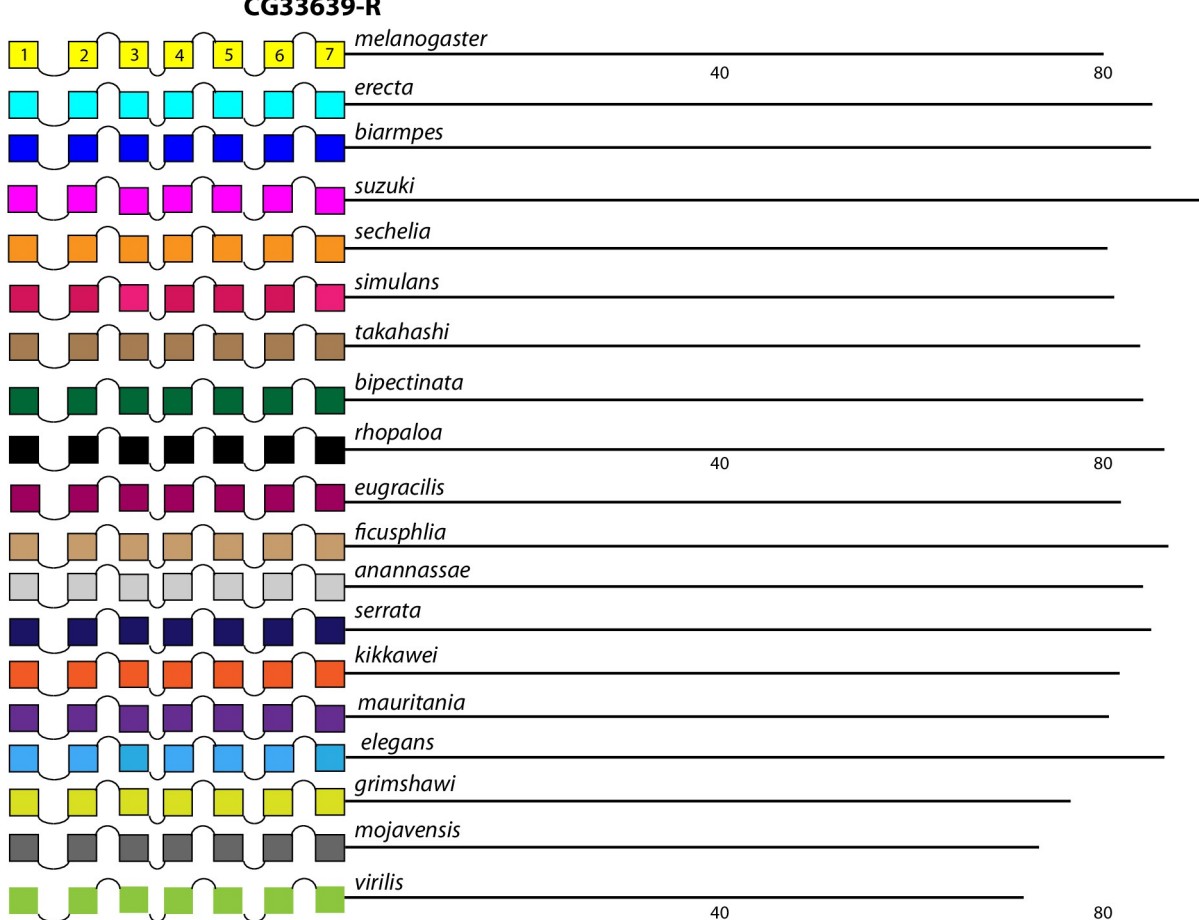

**Fig 20. Extended evolutionary comparison of the GPCR encoded by the *CG33639* gene.** Schematic representations from 19 different *Drosophila* species, including *D. melanogaster* (yellow, top) and *D. virilis* (green, bottom). See Fig 12 Legend for more detail. An initial comparison limited to just *D. melanogaster* and *D. virilis* indicated no conserved BBS sequences in the receptor (Fig 9), and this consideration of an expanded roster of species confirms that conclusion.

conserved BBS sites identify candidate *cis* sequences that regulate the signaling, desensitization and/or trafficking of *Drosophila* neuropeptide GPCRs? Hopefully this work initiates steps towards multiple analyses to help generate useful future hypotheses. Moreover, do the properties of BBS's among different neuropeptide GPCRs (including their number or position) suggest a single mechanism for arrestin-based GPCR regulation, or multiple mechanisms that could differ by cellular or sub-cellular context?

Matches to the complete proposed BBS sequence are present in nearly all 49 *Drosophila* GPCRs examined. Their candidacy as plausible arrestin binding sites was further evaluated by scoring evolutionary durability: that was reported by a comparison of sequences between *D. melanogaster* and *D. virilis* (sub-genera that diverged more than 60 Myr ago). While there is much precedent for assuming that sequence conservation between these two species suggests conserved functional properties, it is also reasonable to ask whether this comparison was in fact too restrictive. I note first that some but not all identified BBS's did not display such conservation, and that therefore the comparison provides a rigorous, initial metric by which to rank candidates for future experimental analyses. Nevertheless, secondary comparisons with species more closely related to *D. melanogaster* revealed that in some cases, the original

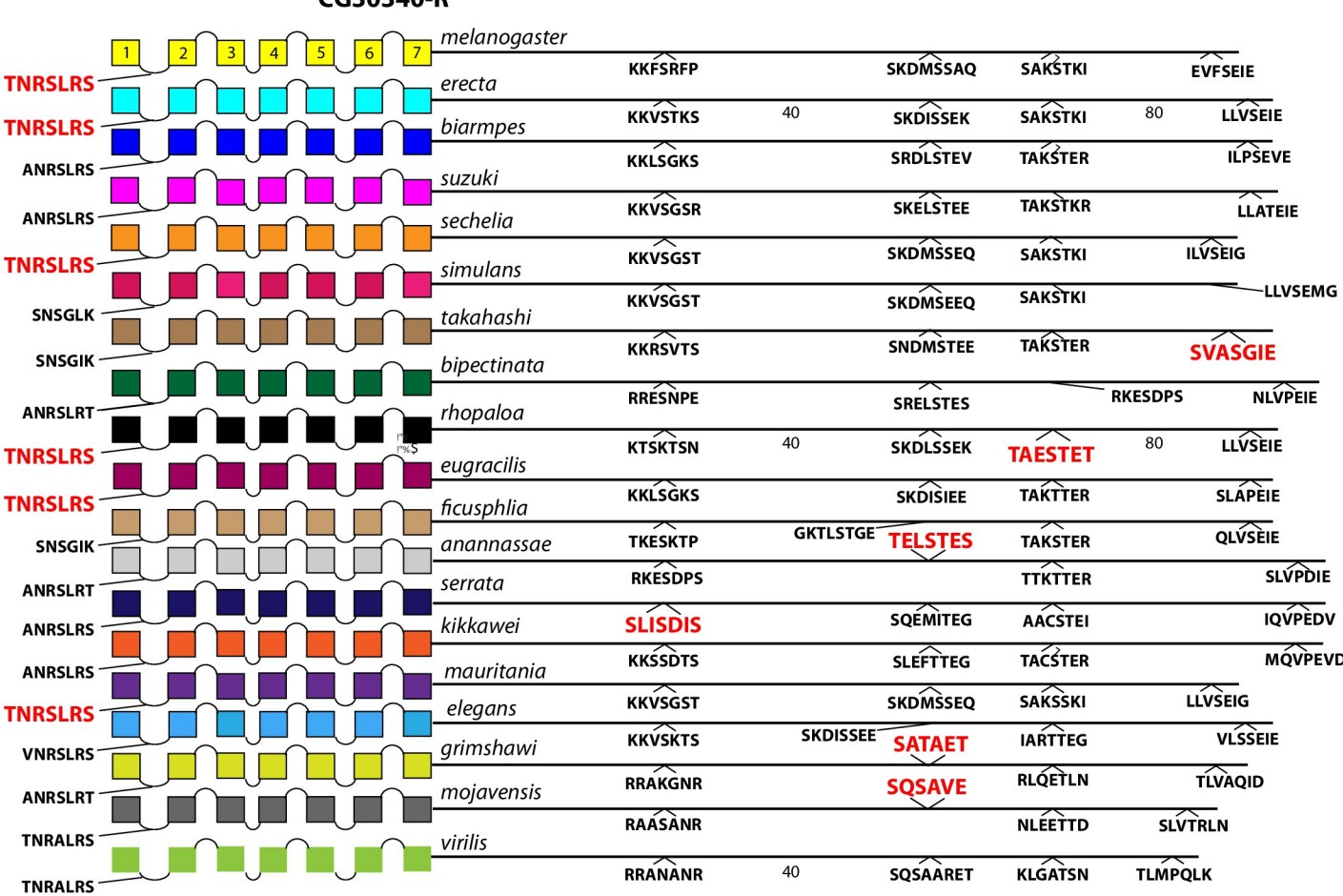

**Fig 21. Extended evolutionary comparison of the orphan GPCR encoded by the *CG30349* gene.** Schematic representations from 19 different *Drosophila* species, including *D. melanogaster* (yellow, top) and *D. virilis* (green, bottom). See Fig 12 Legend for more detail. An initial comparison limited to just *D. melanogaster* and *D. virilis* indicated no conserved BBS sequences in the receptor (Fig 9). Whereas many potential BBS's are found in the expanded roster of species, the over-all lack of sequence conservation supports the initial conclusion.

comparison did indeed produce what appear to be false negatives: many receptors display evolutionarily-conserved BBSs that happen to not extend as far as *D. virilis (*e.g. Proc R, Tk R86C, CAPA R*)*. Thus, there may be good reason to consider further evaluation of those specific *D. melanogaster* BBS sequences in individual GPCRs of interest, despite their lacking conservation in a comparison with *D. virilis*.

## Numbers of putative arrestin binding sites in single *Drosophila* neuropeptide GPCRs

Nearly half of the *Drosophila* neuropeptide GPCRs contain a single conserved BBS in their CT, and about a quarter contain two (Fig 1). This dominant pattern supports a prediction of the model proposed by Zhou *et al.* [1] which features the importance of a single complete BBS. Whereas the majority of *Drosophila* neuropeptide GPCRs display that small number of conserved BBS sites, several notably contain multiple, conserved BBS sequences. The SIFamide R is the extreme case with seven distinct BBS's: together the seven represent ~30% of the nearly ~300 AA CT. The Trissin R and the CCK LR 17D3 likewise contain multiple, long BBS-like sequences with a single ICL domain. How might these more extreme sequence features

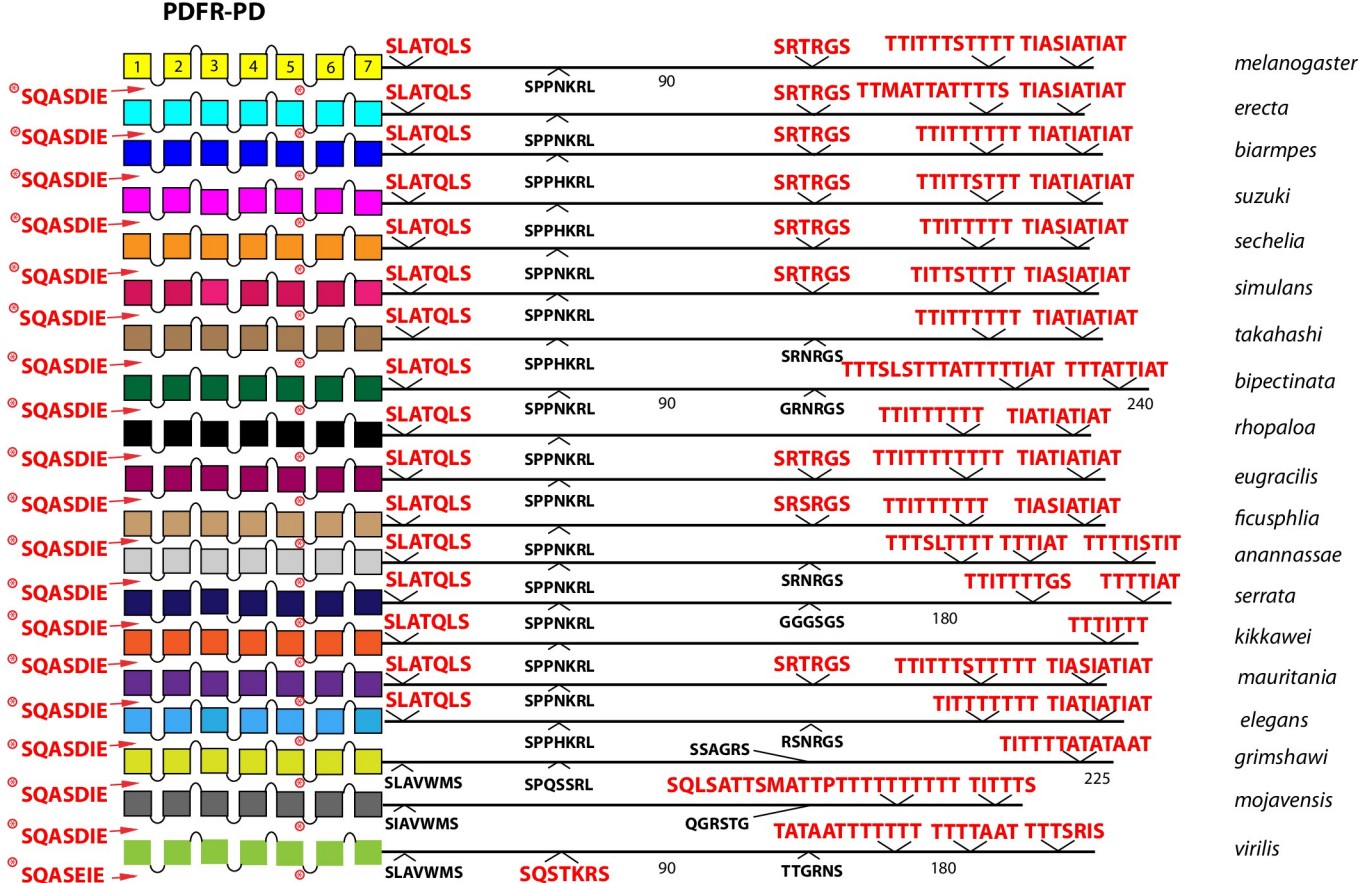

**Fig 22. Extended evolutionary comparison of the PD isoform of the PDF receptor.** Schematic representations from 19 different *Drosophila* species, including *D. melanogaster* (yellow, top) and *D. virilis* (green, bottom). See Fig 12 Legend for more detail. An initial comparison limited to just *D. melanogaster* and *D. virilis* indicated the presence of several clustered BBS sequences in the isoform D-specific CT (Fig 10), and this consideration of an expanded roster of species confirms the conclusion that the D isoform durably encodes a BBS-rich domain.

support GPCR function? Simple speculation posits that different BBS sequences subserve different forms of arrestin regulation. It is well established that distinct patterns of multi-site phosphorylation on a receptor, by different GRK's, differentially enable β-arrestin functions by inducing distinct β-arrestin conformations [46]. This general mechanism has been named the "receptor phosphorylation barcode hypothesis [45, 46] and it emphasizes the importance of the more than one precise pattern of phosphorylated residues on a single GPCR. Alternatively, the presence of multiple, distinct BBS sequences may present a substrate for coupled, hierarchical phosphorylation. Primary phosphorylation of specific sites on the CT of the A3 adenosine receptor [50] and the δ-opioid receptor [51], are required for secondary phosphorylation of residues within ICL3, leading to receptor desensitization. Such a conditional, iterated series of phosphorylation events may underlie sophisticated forms of regulation that feature multiple thresholds, which in turn trigger outcomes that are not distinct, but graded.

## Lengths and locations of putative arrestin binding sites in *Drosophila* neuropeptide GPCRs

A large fraction of the conserved BBS's found in the *Drosophila* neuropeptide GPCRs are longer than the 6–7 AA canonical model first suggested by Zhou *et al.* [1], with the longest being

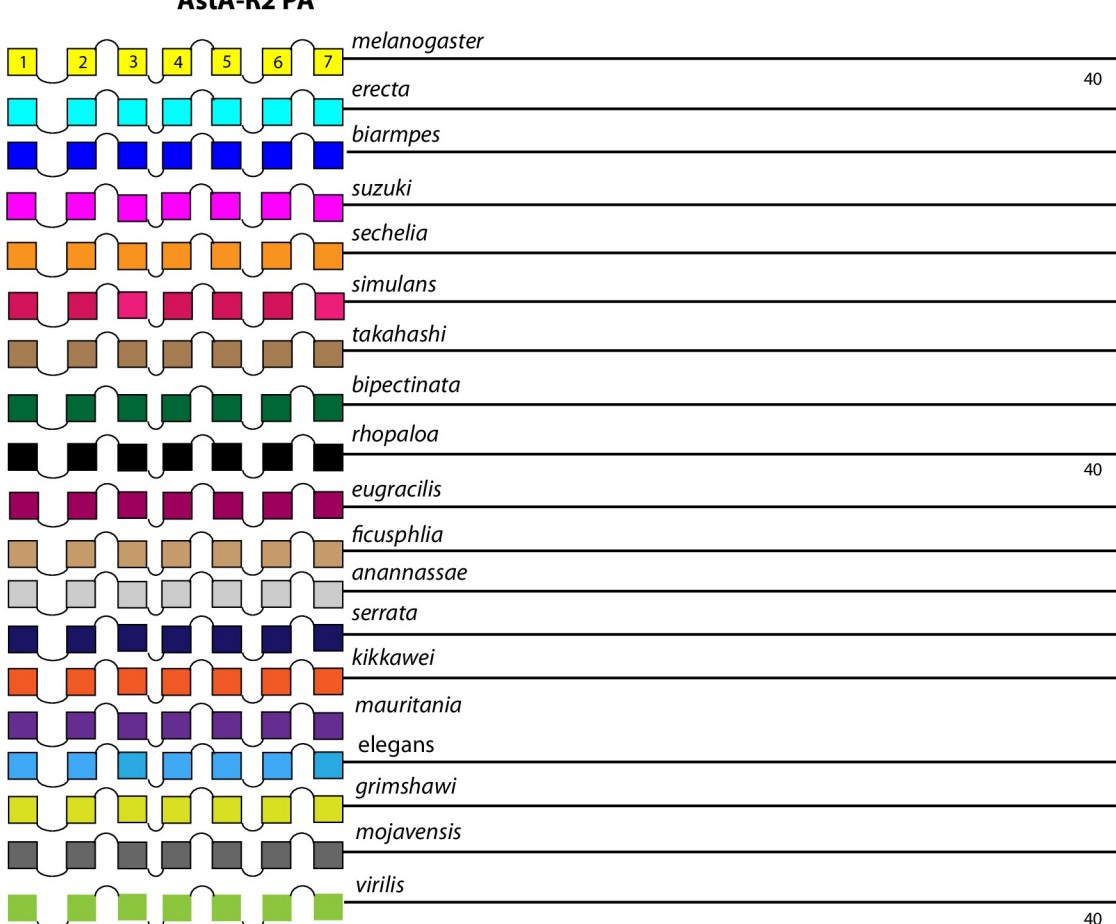

**Fig 23. Extended evolutionary comparison of the PA isoform of the AstA-R2.** Schematic representations from 19 different *Drosophila* species, including *D. melanogaster* (yellow, top) and *D. virilis* (green, bottom). See Fig 12 Legend for more detail. An initial comparison limited to just *D. melanogaster* and *D. virilis* indicated no conserved BBS sequences in the PA isoform and one conserved BBS in the PB isoform (Fig 3). Consideration of the expanded roster of species confirms that conclusion that the absence of BBS sequences is a conserved feature of the AstA R2 PA isoform across the genus *Drosophila*.

20 AAs. It is not yet possible to deduce the significance of these extended, compound BBS sequences, but their conserved character suggests their combination of sequence and length have functional significance. Roughly a third of *Drosophila* neuropeptide GPCRs contain BBS-like sequences in ICL domains. By a large margin, the majority are located in ICL3, a pattern that matches that found in many mammalian GPCRs [e.g., 52–54], and which suggests the importance of precise positions of arrestin-binding sites. One of the *Drosophila* CCK R-like GPCRs includes a 32 AA-long contiguous BBS-like sequence in its ICL3, which represents more than 25% of the ICL3 length. At this stage, there is no obvious explanation for why such an extended compound BBS-like sequence is durably conserved across evolution. A subset of the *Drosophila* GPCRs also contain them in ICL1 and ICL2: Phosphorylation events have been documented on the first and second intracellular loops [46, 54, 55] and in some receptors, ICL2 determinants play an especially important role in regulating arrestin-directed GPCR trafficking [38]. Detailed and systematic genetic analysis *in vivo* of such domains in *Drosophila* may help direct a better understanding of the diversity of lengths and positions of putative arrestin binding sites.

## GPCRs that modify BBS number and sequence by alternative splicing

A number of the neuropeptide GPCRs are expressed as a set of protein isoforms (S1 Table). In certain instances, isoform changes that alter ICL and/ or CT domains create rather precise modifications in BBS number or sequence. The Trissin-R is a prime example in that its three isoforms differ in ways that suggest (but do not prove) there may be biological value in altering the BBS profile to support diverse modes of Trissin-R signaling. The difference between its PB and PD forms represents the use or neglect of the final splice donor site: neglect substitutes for the final 17 AAs of the CT, which includes a 7 AA conserved BBS in PB, with 6 final AAs that lack such in PD. Likewise, in the region encoding a BBS-like sequence in the Trissin-R ICL3, the alternation between two closely-placed splice donor sites creates a subtle change in the properties of the BBS-like sequence in question. In the model postulated by Zhou *et al*. [1], the third acidic residue of the GPCR which promotes alignment to corresponding basic residues of the arrestin, may be either a phosphorylated Ser or Thr, or an acidic AA like Asp or Glu. In the PB and PD isoforms, this BBS-like sequence ends in Asp, while in the PC isoform it presents a Ser. Another change that results from this alternative splicing episode is a substitution of the four AA sequence immediately following the BBS-like position (S2 Fig). It remains possible that this specific and spatially-restricted change of GPCR sequence, which is centered on and alters a conserved BBS-like sequence, in fact has a function separate from arrestin biology. Alternatively, this difference may reveal subtle sequence features of the BBS or adjacent sequences that support differentially-tuned arrestin interactions. The RYa-R also presented differences in isoform expression and degree of BBS sequence conservation (Fig 5). Interestingly, RYa expression is lower in *D. melanogaster* than in *D. virilis* [56] and this difference may be linked to differences in RYa-R isoform BBS sequences and/ or isoform expression.

## GPCRs that lack a BBS

In the examples I here report, five *Drosophila* neuropeptide GPCRs lack conserved BBS's (and several additional neuropeptide GPCRs that produce multiple isoforms include ones that lack such BBS's), supports the possibility that a minority of *Drosophila* neuropeptide GPCRs are desensitized and/ or trafficked by arrestin-independent mechanisms. In this study, I have not included any consideration of partial BBS codes: partial codes contain the first two phosphorylation sites of the complete BBS, but lack the final site [1]. From mutational analysis, Zhou *et al*. [1] concluded that partial BBS sites may support β–arrestin recruitment to certain GPCRs, but posited that high affinity β–arrestin:GPCR interactions generally require the presence of the complete BBS sequence. Hence, I cannot exclude the possibility that some of the *Drosophila* neuropeptide GPCRs that lack a BBS may recruit arrestin interactions via partial sites.

I note however, that there are several well-documented examples of GPCRs that appear to lack robust interactions with arrestins. The β3-adrenergic receptor that does not directly interact with β–arrestins [57] lacks any full or partial phosphorylation code [1]. mGluR5 demonstrates constitutive endocytosis in neurons: it depends on GRK2 to recruit clathrin and thereby undergo desensitization and internalization by a phosphorylation-independent mechanism [58]. Internalization of the β1-adrenergic [59], adrenocorticotropin [60] and leukotriene B4 receptors [61] are all reportedly GRK2-dependent and β-arrestin-independent. Such work suggests that the rates of desensitization and internalization for these GPCRs may depend on endogenous GRK2 levels, which are known to vary across cell types. Furthermore, C1 cannabinoid receptors are normally endocytosed constitutively in hippocampal neurons, in a manner that is not dependent on their activation. They subsequently undergo transcytosis, from somatic-dendritic membrane domains to axonal membrane domains, thereby generating

a polarized cellular expression pattern [62]. Furthermore, it is widely recognized that certain GPCRs are desensitized by heterologous phosphorylation (via PKA or PKC), distinct from homologous modification by GRKs [10]. Hence diverse indications suggest many GPCRs may normally be regulated by β–arrestin-independent mechanisms, and such mechanisms may pertain to that subset of *Drosophila* neuropeptide GPCRs that lack conserved BBS's.

For the large number of GPCRs that associate with β-arrestin, such regulation is significant not only as a mechanism to decrease receptor signaling by de-sensitization and receptor endocytosis [6], but also to traffic and transition GPCRs to later stages of signaling via ERK and other signaling pathways [20, 63]. Studies of β-arrestin regulation of GPCRs in *Drosophila* are less developed than in mammalian models, few in number and scope, and with one notable exception [64], they are limited largely to *in vitro* studies [34, 41–44, 48]. I have reported results here specifically to promote renewed consideration of β-arrestin regulation of neuropeptide GPCRs in *Drosophila* because β-arrestin is arguably central to understanding GPCR signaling, trafficking and turnover. The striking visual differences in the numbers and patterns of highly-conserved BBS profiles across the different fly GPCRs (Figs 3–11) suggest there may exist one or more distinct regulatory paths by which neuropeptide receptor trafficking and signaling proceeds beyond G protein-dependent signaling. However, this conclusion is speculative as such mechanisms are largely unknown in the fly. A sophisticated genetic model system like *Drosophila*, coupled with the rich history of neuroendocrine and neuropeptide physiology in insects [65–67], provides a novel complimentary path forward, for a better understanding of β-arrestin regulatory biology. Given the prominent role of GPCRs in normal signaling, and as targets for a large fraction of modern therapeutics [68, 69], the significance of such model studies could be substantial. Original genetic studies in *Drosophila* helped identify and elucidate many of the major cell signaling pathways in development (e.g., the Notch [70], Hedgehog [71], and decapentaplegic (TGF) [72] pathways, to name but a few). Lastly, because the large majority of *Drosophila* neuropeptide GPCRs derive from ancestors that produced modern mammalian GPCRs [73], the potential is large for mechanisms of neuropeptide GPCR signaling and trafficking revealed in studies of *Drosophila* to have value and impact across a large evolutionary scope.

## Supporting information

**S1 Fig. ICL2 BBS-like sequences.** *Drosophila* Rhodopsin-like GPCRs with BBS-like sequences in ICL2 (bold, italicized) following the conserved Pro (capitalized). No other Rhodopsin-like GPCRs had a BBS-like sequence in either *D. melanogaster* or *D. virilis*.
(PDF)

**S2 Fig. ICL2 domains lacking the canonical Pro at the (DRY)+6 position.** *Drosophila* Rhodopsin-like GPCRs that do not contain a conserved Pro 6 AAs past the <u>DRY</u> sequence of ICL2, instead substituting an Ala or His residue. SP-R has a Pro but it is 7 AAs past the DRY sequence.
(PDF)

**S3 Fig. Alternative splicing that refines the Trissin-R ICL3 BBS-like sequence.** BBS-like and surrounding sequences in the third ICL of Trissin-R PB, PC and PD isoforms. The BBS-like sequence present in the PB and PD isoforms is converted in the PC by alternative splicing (asterisks) yet retains a precise BBS-like character. The correlated change in the surrounding downstream sequence is underlined. Numbers in parentheses identify the predicted AA positions of the first residues illustrated for each isoform (cf. S1 Text and S7 Text).
(PDF)

**S1 Table. Alternative *Drosophila* neuropeptide GPCR CT isoforms.** For each of the 49 neuropeptide GPCRs consider, this table lists the alternative isoforms presented by Flybase ((http://flybase.org/), along with their individual GenBank reference numbers. Isoforms filled in orange indicates they present an alternative CT. [Next column to the right:] The basis for that alternative is indicated as either "alternative splicing" or stop suppression". [Next column to the right:] The number of BBS sequences in the *D. melanogaster* GPCR is followed (/) by the number of those conserved in the *D. virilis* orthologue. [Next column to the right:] The number of BBS-like sequences in the *D. melanogaster* GPCR is followed (/) by the number of those conserved in the *D. virilis* orthologue. [Next column to the right:] the Figure number(s) that illustrate BBS incidence for those GPCRs.
(PDF)

**S1 Text. Collated and annotated *D. melanogaster* and *D. virilis* GPCR sequences.**
(PDF)

**S2 Text. *de novo* [AstA-R2] isoform annotations.**
(PDF)

**S3 Text. *de novo* [CCAP-R] isoform annotations.**
(PDF)

**S4 Text. *de novo* [RYa-R] isoform annotations.**
(PDF)

**S5 Text. *de novo* [Tk-R 86C] isoform annotations.**
(PDF)

**S6 Text. *de novo* [CAPA-R] isoform annotations.**
(PDF)

**S7 Text. *de novo* [Trissin-R] isoform annotations.**
(PDF)

**S8 Text. *de novo* [Moody] isoform annotations.**
(PDF)

**S9 Text. Multi-species analysis of Proc-R.**
(PDF)

**S10 Text. Multi-species analysis of Tk-R 86C.**
(PDF)

**S11 Text. Multi-species analysis of CAPA-R.**
(PDF)

**S12 Text. Multi-species analysis of PDF-R PAs.**
(PDF)

**S13 Text. Multi-species analysis of CCHa-R2.**
(PDF)

**S14 Text. Multi-species analysis of LGR1.**
(PDF)

**S15 Text. Multi-species analysis of NPF-R.**
(PDF)

**S16 Text. Multi-species analysis of Hector-R.**
(PDF)

**S17 Text. Multi-species analysis of CG33639-R.**
(PDF)

**S18 Text. Multi-species analysis of CG30340-R.**
(PDF)

**S19 Text. Multi-species analysis of PDF-R PDs.**
(PDF)

**S20 Text. Multi-species analysis of AstA-R2 PAs.**
(PDF)

## Acknowledgments

PHT thanks Flybase (http://flybase.org/) for providing and maintaining a useful database that was instrumental in supporting pursuit of this work.

## Author Contributions

**Data curation:** Paul H. Taghert.

**Formal analysis:** Paul H. Taghert.

**Funding acquisition:** Paul H. Taghert.

**Validation:** Paul H. Taghert.

**Visualization:** Paul H. Taghert.

**Writing – original draft:** Paul H. Taghert.

**Writing – review & editing:** Paul H. Taghert.

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
