## [Decision Letter · Decision Letter 0]

6 Oct 2022

PONE-D-22-25817The incidence of candidate binding sites for beta-arrestin in *Drosophila* neuropeptide GPCRs PLOS ONE

Dear Dr. Taghert,

Thank you for submitting your manuscript to PLOS ONE. After careful consideration, we feel that it has merit but does not fully meet PLOS ONE’s publication criteria as it currently stands. Therefore, we invite you to submit a revised version of the manuscript that addresses the points raised during the review process.

Please submit your revised manuscript by Nov 20 2022 11:59PM. If you will need more time than this to complete your revisions, please reply to this message or contact the journal office at plosone@plos.org. Please include the following items when submitting your revised manuscript:A rebuttal letter that responds to each point raised by the academic editor and reviewer(s). You should upload this letter as a separate file labeled 'Response to Reviewers'.A marked-up copy of your manuscript that highlights changes made to the original version. You should upload this as a separate file labeled 'Revised Manuscript with Track Changes'.An unmarked version of your revised paper without tracked changes. You should upload this as a separate file labeled 'Manuscript'.If applicable, we recommend that you deposit your laboratory protocols in protocols.io to enhance the reproducibility of your results. Protocols.io assigns your protocol its own identifier (DOI) so that it can be cited independently in the future. For instructions see: https://journals.plos.org/plosone/s/submission-guidelines#loc-laboratory-protocols. Additionally, PLOS ONE offers an option for publishing peer-reviewed Lab Protocol articles, which describe protocols hosted on protocols.io. Read more information on sharing protocols at https://plos.org/protocols?utm_medium=editorial-email&utm_source=authorletters&utm_campaign=protocols.

We look forward to receiving your revised manuscript.

Kind regards,

Guangyu Wu, PhD

Academic Editor

PLOS ONE

Journal Requirements:

3. Please expand the acronym “NIH - NINDS ” (as indicated in your financial disclosure) so that it states the name of your funders in full.

Reviewers' comments:

Reviewer's Responses to Questions

**Comments to the Author**

1. Is the manuscript technically sound, and do the data support the conclusions?

Reviewer #1: Yes

Reviewer #2: Yes

2. Has the statistical analysis been performed appropriately and rigorously? 

Reviewer #1: Yes

Reviewer #2: N/A

3. Have the authors made all data underlying the findings in their manuscript fully available?

Reviewer #1: Yes

Reviewer #2: Yes

4. Is the manuscript presented in an intelligible fashion and written in standard English?

Reviewer #1: Yes

Reviewer #2: Yes

5. Review Comments to the Author

**Reviewer #1:** This is an admirable attempt to understand which amino acid residues of Drosophila neuropeptide GPCRs may be involved in binding to beta-arrestin. As the author states it is the initial identification of putative arrestin binding sites in these receptors, a prerequisite for before an experimental exploration of this problem can be envisaged. It is a very thorough and complete description of putative beta-arrestin binding sites (BBS's) in Drosophila neuropeptide GPCRs. Many of the putative BBS's are well conserved and have thus to be physiologically relevant. It is an impressive piece of work both in its thoroughness, which required in several instances to correct sequences in the data bank, as well as its completeness.

I have only a limited number of comments which the author may consider for improving this interesting manuscript.

****Comparing D. virilis GPCRs with those from D. melanogaster is valid approach, however one might expect that if the expresssion of a neuropeptide is signficantly different between these two species, its receptors might be different as well. As far as we know, for most neuropeptides expression in these two species seems quite similar, however that of RYamide is remarkably different. In D. melanogaster the expression of RYamide is much reduced as compared to other Drosophila species. This may help to explain the apparent differences between two of the three RYamide receptors from D.virilis and D. melanogaster. For this reason, a figure for the three RYamide receptors similar to the ones done in Figures 12-23 might be useful.

“complete” site as defined by 1. Zhou et al. (1).

****this should be “complete” site as defined by Zhou et al. (1).

"I judged orthologous BBS sites by proximity (< 20 bp distance, as bounded by the end of the 7 th TM domain and the end of the CT),"

****I guess 20 bp should have been amino acid residues.

****It is impossible to predict of course whether all these BBS's are indeed functional. One question that comes to mind is what is the probability to find a complete site [S/T-(X1)-S/T-(X2)-(X3)-(S/T/E/D)] in a random protein sequence. As a very crude, and most likely incorrect, approximation I used the number of codon exons to estimate the occurence of amino acids. S/T p=10/64 (X1), (X2), (X3), almost 1, (S/T/E/D) p=14/64. So such sequences can be expected to be reasonably common just by chance, roughly one occurence in every 200 amino acid residues. A similar chance for the 7 amino acid residue form, suggest either a short or long site ever 100 amino acid residues. Thus the number of putative BBS's does not seem to be that different from what might be expected by chance alone. Of course this does not mean that this sequence would not be functional.

"Secondly, the majority of BBS’s were either 6 or 7 AAs in length (Figure 1B)".

****This sentence surprised me, as the author previously defined BBS's as either having 6 or 7 AAs. Looking at the figures, it becomes clear that some sequences contain multiple subsequences of BBS's. This could perhaps be indicated when describing the BBS's, something like "some BBS's overlap and yield larger...". Within this context my gut feeling, a very unreliable signal, tells me that some of the multiple T/S sequences are unlikely arestin binding sites.

****Fig. 2B. This is a minor comment. The scale of the D. virilis is different from the one in D. melanogaster, which makes it more difficulte to compare the two. Making the scale in D. virilis the same size, and thus increase the size of the figure, this would become clearer and since there is space in Figure 2 to do so, the author might want to consider that.

"Lastly, and in contrast to the situation for CTs, there was a prominent non-random disposition of BBS-like sequences among ICL domains, with the strongest representation in ICL3 (Figure 2C)".

****If ICL1 is much shorter e.g. than the non-random disposition might be due to a non-random distribution of length of the ICLs.

"The BBS in the D. virilis AstA R2 PB does not maintain sequence conservation, but I rated it conserved based on its precisely correspondent position within the final alternative exon."

****I guess this will be an interesting one for experimental validation, as will be some of those that are present in D. virilis but not D. melanogaster or vice versa. This is the type of information that will be useful for further study.

**Reviewer #2:** The manuscript by Taghert describes a careful and comprehensive bioinformatic analysis of predicted β-arrestin binding sites across the peptide GPCRs of Drosophila melanogaster, and in comparison other Drosophila-species. The analysis is based on existing genomic and transcriptomic sequence information available in FlyBase and is purely theoretical. Yet, it produces a wealth of highly interesting data (one could also say a catalagoue for predicted β-arrestin binding sites) for a whole “receptome” and likely will serve as a solid platform for many future experimental studies on the regulatory mechanisms underlying GPCR activity. The fruit fly Drosophila with its unmatched genetic tools that allow such studies appears very well chosen. As such, I consider it a highly significant and original piece of work.

The manuscript is well and carefully written, and results are presented in 23 clearly laid out graphs plus 3 supplementary graphs and detailed sequence annotations/analyses in tables and text. The online format of P1 seems well suited for this quite extensive documentation, and I suggest to not cut the number of graphs. I have no major issues with any manuscript sections, yet some minor comments (see below). I have done this before, but are happy to repeat: it would massively help reviewers of P1 manuscripts if the manuscripts had page numbers and even better line numbers.

- General: please unify the nomenclature (best perhaps following FlyBase). Sometimes, receptors are written as XY-R, sometimes its XY R.

-Introduction, first paragraph: can the term “Heterologous” be better explained? Is this restricted to PKA/PKC phosphorylation upon activation of a different GPCR than the one to be phosphorylated?

-Introduction, second page first paragraph: “Strong affinity is correlated with sustain interaction..” should be “sustained” I guess

-Results, second page: “….BBS from was also proposed, by which the first and third S/T residues….” Do you refer to the position (S/T-X1-X2-S/T), as there is a second S/T residue inbetween the first and third S/T. Please clarify.

- Results, fifth page: “.. based on conformation features presented several distinc regions, including..” and “..these correspondent (and hence conserved) based their position…”

Please check grammar, I find it impossible to understand.

-Results, sixth page, first line: Of the 19 GPCRs…17 contained… three of these 13… Three other GPCRs…

The numbers don’t seem to match (e.g. 17+3 =20) or I simply don’t get it – correct or rewrite for people like me.

-Results, sixth page, bottom: “..AstAR1 also has A paralogous receptor..” I suggest to give the name (AstA R2 I guess) for the sake of clarity.

-Results, seventh page: “…GPCR sequences which ARE found….

-Results, eighteenth page and ff.: “Drosophila suzuki” should be “Drosophila suzukii”

-Discussion, sixth page, end of first paragraph: “..some of the many Drosophila neuropeptide GPCRs..”

“many” seems somewhat inappropriate. If I count correctly, it’s just five of them(?)

-Discussion, sixth page, second paragraph:

“The b3-adrenergic receptor…” should be greek letter beta

“mGlur5 demonstrates…” should be “mGluR5”

-Discussion, seventh page, second paragraph: “..there may exist a one or more distinct regulatory paths…” correct grammar

6. PLOS authors have the option to publish the peer review history of their article (what does this mean?). If published, this will include your full peer review and any attached files.

Reviewer #1: No

Reviewer #2: No

---

## [Author Response · Author response to Decision Letter 0]

10 Oct 2022

I have made appropriate revisions.

I apologize for this confusion; The work was funded by a NIH grant from the National Institute of Neurological Disorders and Stroke

• 3. Please expand the acronym “NIH - NINDS ” (as indicated in your financial disclosure) so that it states the name of your funders in full.

thank you

• 4. Please review your reference list to ensure that it is complete and correct. 

I added a reference based on a comment by one of the reviewers (#56 in the revised manuscript).

Here follows the point-by-point response to the reviews:

• Reviewer #1: This is an admirable attempt to understand which amino acid residues of Drosophila neuropeptide GPCRs may be involved in binding to beta-arrestin. As the author states it is the initial identification of putative arrestin binding sites in these receptors, a prerequisite for before an experimental exploration of this problem can be envisaged. It is a very thorough and complete description of putative beta-arrestin binding sites (BBS's) in Drosophila neuropeptide GPCRs. Many of the putative BBS's are well conserved and have thus to be physiologically relevant. It is an impressive piece of work both in its thoroughness, which required in several instances to correct sequences in the data bank, as well as its completeness.

• I have only a limited number of comments which the author may consider for improving this interesting manuscript.

****Comparing D. virilis GPCRs with those from D. melanogaster is valid approach, however one might expect that if the expresssion of a neuropeptide is signficantly different between these two species, its receptors might be different as well. As far as we know, for most neuropeptides expression in these two species seems quite similar, however that of RYamide is remarkably different. In D. melanogaster the expression of RYamide is much reduced as compared to other Drosophila species. This may help to explain the apparent differences between two of the three RYamide receptors from D.virilis and D. melanogaster. For this reason, a figure for the three RYamide receptors similar to the ones done in Figures 12-23 might be useful.

• This is an interesting observation; I will decline the invitation to produce a pan-Sophophora Figure for the RYa-Rs, because the PB isoforms have to be reconstructed by manual analysis. I have revised the Discussion of alternative GPCR splicing (line 767) to include mention of the reviewer’s suggestion, as follows:

“The RYa R also presented differences in isoform expression and degree of BBS sequence conservation (Figure 5). Interestingly, RYa expression is lower in D. melanogaster than in D. virilis (56) and this difference may be linked to differences in RYa-R receptor isoform BBS sequences and/ or isoform expression.”

• 

“complete” site as defined by 1. Zhou et al. (1).

****this should be “complete” site as defined by Zhou et al. (1).

Thank you – Corrected, as suggested.

• 

"I judged orthologous BBS sites by proximity (< 20 bp distance, as bounded by the end of the 7 th TM domain and the end of the CT),"

****I guess 20 bp should have been amino acid residues.

• Thank you – Corrected, as suggested. 

****It is impossible to predict of course whether all these BBS's are indeed functional. One question that comes to mind is what is the probability to find a complete site [S/T-(X1)-S/T-(X2)-(X3)-(S/T/E/D)] in a random protein sequence. As a very crude, and most likely incorrect, approximation I used the number of codon exons to estimate the occurrence of amino acids. S/T p=10/64 (X1), (X2), (X3), almost 1, (S/T/E/D) p=14/64. So such sequences can be expected to be reasonably common just by chance, roughly one occurrence in every 200 amino acid residues. A similar chance for the 7 amino acid residue form, suggest either a short or long site ever 100 amino acid residues. Thus the number of putative BBS's does not seem to be that different from what might be expected by chance alone. Of course this does not mean that this sequence would not be functional.

• The reviewer makes an excellent point – the features of a match to the model BBS site are not so unusual and have a reasonable probability of occurrence by chance. For this reason, I have framed considerations of potential functionality of BBS sites based on evolutionary conservation.

"Secondly, the majority of BBS’s were either 6 or 7 AAs in length (Figure 1B)".

****This sentence surprised me, as the author previously defined BBS's as either having 6 or 7 AAs. Looking at the figures, it becomes clear that some sequences contain multiple subsequences of BBS's. This could perhaps be indicated when describing the BBS's, something like "some BBS's overlap and yield larger...". Within this context my gut feeling, a very unreliable signal, tells me that some of the multiple T/S sequences are unlikely arestin binding sites.

• Thank you for this suggestion: I modified the text to read: “About 1/3 of the BBS’s were longer (>10 AA) and contained multiple BBSs (these represent compound sites which, as noted by Zhou et al. (1) occurs in the rhodopsin CT,…”

****Fig. 2B. This is a minor comment. The scale of the D. virilis is different from the one in D. melanogaster, which makes it more difficulte to compare the two. Making the scale in D. virilis the same size, and thus increase the size of the figure, this would become clearer and since there is space in Figure 2 to do so, the author might want to consider that.

• This is an interesting suggestion, but I prefer to leave it as presented. This point is in fact discussed in the text. So I don’t think the interested reader will be led astray by the Figure.

"Lastly, and in contrast to the situation for CTs, there was a prominent non-random disposition of BBS-like sequences among ICL domains, with the strongest representation in ICL3 (Figure 2C)".

****If ICL1 is much shorter e.g. than the non-random disposition might be due to a non-random distribution of length of the ICLs.

• This is a reasonable point as I had not taken the length of ICL regions into account. I therefore revised the text to read: “Lastly, among ICL domains, BBS-like sequences were most strongly represented in ICL3 (Figure 2C).”

"The BBS in the D. virilis AstA R2 PB does not maintain sequence conservation, but I rated it conserved based on its precisely correspondent position within the final alternative exon."

****I guess this will be an interesting one for experimental validation, as will be some of those that are present in D. virilis but not D. melanogaster or vice versa. This is the type of information that will be useful for further study.

• Agreed.

• Reviewer #2: The manuscript by Taghert describes a careful and comprehensive bioinformatic analysis of predicted β-arrestin binding sites across the peptide GPCRs of Drosophila melanogaster, and in comparison other Drosophila-species. The analysis is based on existing genomic and transcriptomic sequence information available in FlyBase and is purely theoretical. Yet, it produces a wealth of highly interesting data (one could also say a catalagoue for predicted β-arrestin binding sites) for a whole “receptome” and likely will serve as a solid platform for many future experimental studies on the regulatory mechanisms underlying GPCR activity. The fruit fly Drosophila with its unmatched genetic tools that allow such studies appears very well chosen. As such, I consider it a highly significant and original piece of work.

The manuscript is well and carefully written, and results are presented in 23 clearly laid out graphs plus 3 supplementary graphs and detailed sequence annotations/analyses in tables and text. The online format of P1 seems well suited for this quite extensive documentation, and I suggest to not cut the number of graphs. I have no major issues with any manuscript sections, yet some minor comments (see below). I have done this before, but are happy to repeat: it would massively help reviewers of P1 manuscripts if the manuscripts had page numbers and even better line numbers.

Apologies - Have now added both

General: please unify the nomenclature (best perhaps following FlyBase). Sometimes, receptors are written as XY-R, sometimes its XY R.

Thank you for this suggestion. I now maintain an X-R format throughout.

Introduction, first paragraph: can the term “Heterologous” be better explained? Is this restricted to PKA/PKC phosphorylation upon activation of a different GPCR than the one to be phosphorylated?

Thanks for this question. As suggested, desensitization follows activation of a different receptor. Mechanisms so far defined include the participation by PKA or PKC. My feeling is that because it is not a main consideration of this manuscript, it did not warrant detailed explanation. Still interested readers can learn more by studying indicated citation.

Introduction, second page first paragraph: “Strong affinity is correlated with sustain interaction..” should be “sustained” I guess

Thank you – now corrected

Results, second page: “….BBS from was also proposed, by which the first and third S/T residues….” Do you refer to the position (S/T-X1-X2-S/T), as there is a second S/T residue inbetween the first and third S/T. Please clarify.

Thank you for identifying this mistake – the text was revised to read: “A long” BBS form was also proposed, by which the second and third S/T residues are separated by a second X, with no restricted AAs. 

Results, fifth page: “.. based on conformation features presented several distinc regions, including..” and “..these correspondent (and hence conserved) based their position…”

Please check grammar, I find it impossible to understand.

Thank you for these suggestions. I have revised the text to read:

“Direct cryo-EM observations reveal details of �-arrestin interactions with the receptor core (36). There is strong evidence that �-arrestin interacts with GPCRs based on conformational features presented several distinct regions, including intracellular domains that are independent of the CT region (37).”

“The reason for the discrepancy was a single example (CCKL-R 17D3) in which a single long BBS-like in D. virilis is divided by two non-complying AAs in D. melanogaster. Based on their proximity and over-all sequence similarities, I rated these two nearly-contiguous BBS sites in D. melanogaster as correspondent to the single one in D. virilis (and hence conserved): these are detailed in later Figures”

-Results, sixth page, first line: Of the 19 GPCRs…17 contained… three of these 13… Three other GPCRs…

The numbers don’t seem to match (e.g. 17+3 =20) or I simply don’t get it – correct or rewrite for people like me.

Thank you for pointing out my error. The text was revised to read as follows:

“Of the 19 GPCRs containing BBS-like sequences, 17 contained one or more in ICL3: three of these 17 receptors also contained them in ICL1, and two contained them in ICL2. Three other GPCRs contained conserved BBS-like sequences in ICL2 only.”

Results, sixth page, bottom: “..AstAR1 also has A paralogous receptor..” I suggest to give the name (AstA R2 I guess) for the sake of clarity.

The text was revised as follows:

“AstA-R1 also has paralogous receptor, AstA-R2, but the latter does not contain an ICL2 BBS-like sequence.”

Results, seventh page: “…GPCR sequences which ARE found….

Thank you - the word ‘are’ was added.

Results, eighteenth page and ff.: “Drosophila suzuki” should be “Drosophila suzukii”

• Corrected – thank you.

Discussion, sixth page, end of first paragraph: “..some of the many Drosophila neuropeptide GPCRs..”

“many” seems somewhat inappropriate. If I count correctly, it’s just five of them(?)

Thank you – I have deleted the word ‘many’.

Discussion, sixth page, second paragraph:

“The b3-adrenergic receptor…” should be greek letter beta

Corrected – thank you

“mGlur5 demonstrates…” should be “mGluR5”

Corrected – thank you

-Discussion, seventh page, second paragraph: “..there may exist a one or more distinct regulatory paths…” correct grammar

Corrected – thank you

---

## [Editor Report · Decision Letter 1]

17 Oct 2022

The incidence of candidate binding sites for beta-arrestin in *Drosophila* neuropeptide GPCRs 

PONE-D-22-25817R1

Dear Dr. Taghert ,

We’re pleased to inform you that your manuscript has been judged scientifically suitable for publication and will be formally accepted for publication once it meets all outstanding technical requirements.

Kind regards,

Guangyu Wu, PhD

Academic Editor

PLOS ONE

---

## [Editor Report · Acceptance letter]

21 Oct 2022

PONE-D-22-25817R1 

The incidence of candidate binding sites for b-arrestin in Drosophila neuropeptide GPCRs 

Dear Dr. Taghert:

I'm pleased to inform you that your manuscript has been deemed suitable for publication in PLOS ONE. Congratulations! Your manuscript is now with our production department. 

Kind regards, 

on behalf of

Dr. Guangyu Wu 

Academic Editor

PLOS ONE